



# Influence of low-level blocking and turbulence on the microphysics of a mixed-phase cloud in an inner-Alpine valley

Fabiola Ramelli[1], Jan Henneberger[1], Robert O. David[2], Annika Lauber[1], Julie T. Pasquier[1], Jörg Wieder[1], Johannes Bühl[3], Patric Seifert[3], Ronny Engelmann[3], Maxime Hervo[4], and Ulrike Lohmann[1]

[1]Department of Environmental System Sciences, Institute for Atmospheric and Climate Science, ETH Zurich, Zurich, Switzerland
[2]Department of Geosciences, University of Oslo, Oslo, Norway
[3]Leibniz Institute for Tropospheric Research, Leipzig, Germany
[4]Federal Office of Meteorology and Climatology MeteoSwiss, Payerne, Switzerland

**Correspondence:** Fabiola Ramelli (fabiola.ramelli@env.ethz.ch) and Jan Henneberger (jan.henneberger@env.ethz.ch)

**Abstract.** Previous studies that investigated orographic precipitation have primarily focused on isolated mountain barriers. Here we investigate the influence of low-level blocking and shear-induced turbulence on the cloud microphysics and precipitation formation in a complex inner-Alpine valley. The analysis focuses on a mid-level cloud in a post-frontal environment, by combining observations from an extensive set of instruments including ground-based remote sensing instrumentation, in situ
5   instrumentation on a tethered balloon system and ground-based precipitation measurements.

During this event, the boundary layer was characterized by a blocked low-level flow and a turbulent shear layer, which separated the blocked layer near the surface from the stronger cross-barrier flow aloft. Cloud radar observations indicate changes in the microphysical cloud properties within the turbulent shear layer including enhanced linear depolarization ratio (i.e., change in particle shape) and increased radar reflectivity (i.e., enhanced ice growth). Based on the ice particle habits observed at the
10   surface, we suggest that needle growth and aggregation occurred within the turbulent layer and that collisions of fragile ice crystals (e.g., dendrites, needles) might have contributed to secondary ice production.

Additionally, in situ instrumentation on the tethered balloon system observed the presence of a low-level feeder cloud above a small-scale topographic feature, which dissipated when the low-level flow turned from a blocked to an unblocked state. Our observations indicate that the low-level blocking (due to the downstream mountain barrier) caused the low-level flow to ascend
15   the leeward slope of the local topography in the valley, thus producing a low-level feeder cloud. Although the feeder cloud did not enhance precipitation in the present case, we propose that local flow effects such as low-level blocking can induce the formation of feeder clouds in mountain valleys and on the leeward slope of foothills upstream of the main mountain barrier, where they can act to enhance orographic precipitation through the seeder-feeder mechanism.





## 1 Introduction

Mountains can alter and reorganize incoming weather systems or force air masses to lift and thus produce a large proportion of the Earth's annual precipitation (Roe, 2005). Besides the total amount of precipitation, also its spatial distribution across the mountain range becomes of increasing importance for public warning (e.g., avalanche, flash flood), water resources, hydropower production and winter tourism (Stoelinga et al., 2013). In addition to orographic lifting and the subsequent production of condensate, additional processes are required to efficiently form precipitation-sized particles within the lifetime of the cloud as it crosses the mountain barrier (e.g., Smith, 1979; Frei and Schär, 1998; Roe, 2005; Houze Jr, 2012; Smith, 2019). As such, an extensive knowledge of these physical processes and the interplay between dynamics, microphysics and orography is essential to understand precipitation formation over complex terrain.

Numerous mechanisms have been identified to affect the air flow and enhance orographic precipitation (e.g., Borys et al., 2003; Rotunno and Houze, 2007; Lowenthal et al., 2011; Houze Jr, 2012; Medina and Houze Jr, 2015; Kirshbaum et al., 2018; Smith, 2019). For example, if the air flow impinging on a mountain barrier is sufficiently weak, or the mountain barrier is too high or the atmosphere is stably stratified, the low-level flow might be blocked or diverted around the mountain. As a consequence, a stagnant blocked layer can form in front of the mountain barrier, which extends the effective width of the mountain barrier and causes lifting further upstream (e.g. Rotunno and Ferretti, 2001; Medina and Houze, 2003; Jiang and Smith, 2003). Additionally, a layer of strong shear is usually present at the interface between the blocked layer and the strong cross-barrier flow aloft, which can be the source of turbulent motions. This shear-induced turbulent layer can enhance orographic precipitation (e.g. Houze Jr and Medina, 2005; Medina et al., 2005; Medina et al., 2007) and will be the focus of the present study.

Besides the dynamical response of the air flow to the orography, a wide range of microphysical interactions can occur between cloud droplets, ice crystals and water vapor. For example, individual ice crystals can grow by vapor deposition, can collide and stick together with other ice crystals (aggregation) or can collide with supercooled cloud droplets that freeze upon contact (riming) (e.g. Pruppacher and Klett, 1980; Lohmann et al., 2016). Turbulence and updrafts can accelerate ice growth by riming and aggregation and thus precipitation fallout, by sustaining the production of supercooled liquid water (Rauber and Tokay, 1991) and by increasing the collision efficiencies between cloud particles (Pinsky et al., 2016). Furthermore, enhanced ice-ice collisions can promote mechanical break-up of ice crystals and lead to the production of a large number of small secondary ice particles (e.g., Vardiman, 1978; Yano et al., 2016).

In the present study, we investigate the influence of low-level blocking and shear-induced turbulence on the microphysics and precipitation formation of a mixed-phase cloud in an inner-Alpine valley. Previous studies found that flow blocking and shear-induced turbulence can facilitate rapid ice growth and ultimately enhance orographic precipitation (e.g., Marwitz, 1983; Overland and Bond, 1995; Yu and Smull, 2000; Hogan et al., 2002; Neiman et al., 2002; Neiman et al., 2004; Houze Jr and Medina, 2005; Loescher et al., 2006; Olson et al., 2007; Olson and Colle, 2009; Geerts et al., 2011; Medina and Houze Jr, 2015; Grazioli et al., 2015; Aikins et al., 2016). For example, turbulent updraft cells were observed over the Oregon Cascade Mountains in regions where the wind shear exceeded $10\,\mathrm{m\,s^{-1}\,km^{-1}}$ (Houze Jr and Medina, 2005). Houze Jr and Medina (2005) suggested that these turbulent updraft cells can enhance ice growth through riming and aggregation. Both mechanisms can lead





to rapid conversion of condensate to precipitation-sized particles. More recent studies confirmed the findings by Houze Jr and
Medina (2005) that turbulent updraft cells enhance ice growth and precipitation (e.g., Medina and Houze Jr, 2015; Geerts et al.,
2011; Aikins et al., 2016). However, Geerts et al. (2011) could not draw any conclusions regarding the dominant ice growth
processes, due to restrictions in the aircraft flight level. On the other hand, Aikins et al. (2016) proposed depositional growth
and aggregation as the dominant ice growth mechanisms for their study rather than riming due to the low amounts of liquid
water observed in the shear layer. Thus, the dominant growth process within shear-induced turbulent layers depends on the
environmental conditions such as temperature, updraft velocity and the ice crystal size distribution.

The present work builds on the previous studies that investigated the implications of shear-induced turbulence on the cloud
microphysics and precipitation formation and extends the analysis to a more complex terrain. While previous observational
studies have mainly focused on the effect of an isolated mountain barrier, we will investigate the role of shear-induced tur-
bulence in an inner-Alpine valley near Davos, Switzerland. The region around Davos, or more generally the Alpine region,
is characterized by complex terrain with narrow valleys and multiple mountain barriers, which can cause complex interac-
tions between numerous mechanisms on different scales. This complexity has already been recognized during the Mesoscale
Alpine Programme (MAP) (e.g., Rotunno and Houze, 2007). In an environment with a series of parallel mountain ridges, a
superposition of upstream and downstream effects can occur. In the present study, we attempt to investigate whether a 'simple'
conceptual mechanism as described in Houze Jr and Medina (2005) can also be observed in a complex environment that is
embedded between two mountain ridges and which is likely influenced by upstream and downstream effects. In addition, we
extend the analysis to lower altitudes, which were inaccessible in previous studies due to limits in the flight levels, by using
a tethered balloon system. The balloon-borne profiles can provide information on the microphysical cloud properties in the
lowest part of the boundary layer. These questions are addressed in a case study on 7 March 2019, which was observed in a
post-frontal air mass during the Role of Aerosols and CLouds Enhanced by Topography on Snow campaign (RACLETS). The
multi-instrument analysis is based on (1) wind profiler and wind lidar measurements, (2) observations of a Ka-band polari-
metric cloud radar, a Raman lidar and a microwave radiometer, (3) in situ microphysical measurements on a tethered balloon
system and (4) ground-based precipitation measurements.

The main measurement locations and instruments are briefly described in Section 2. An overview of the synoptic weather
situation and the case study is given in Section 3. Section 4 presents the influence of low-level blocking and shear-induced
turbulence on the cloud microphysics and precipitation formation. The findings are discussed in a larger context and presented
in a conceptual model in Section 5. A summary of the main findings is given in Section 6.

## 2 Measurement location and instruments

The RACLETS campaign took place from 8 February 2019 to 28 March 2019 in the region around Davos in the Swiss Alps.
The main objective of the campaign was to investigate the pathways of precipitation formation in orographic clouds, covering
the entire aerosol-cloud-precipitation-snow distribution process chain, in order to improve our understanding of orographic
precipitation. The overall goal of the RACLETS campaign was to use the gained process understanding of orographic clouds




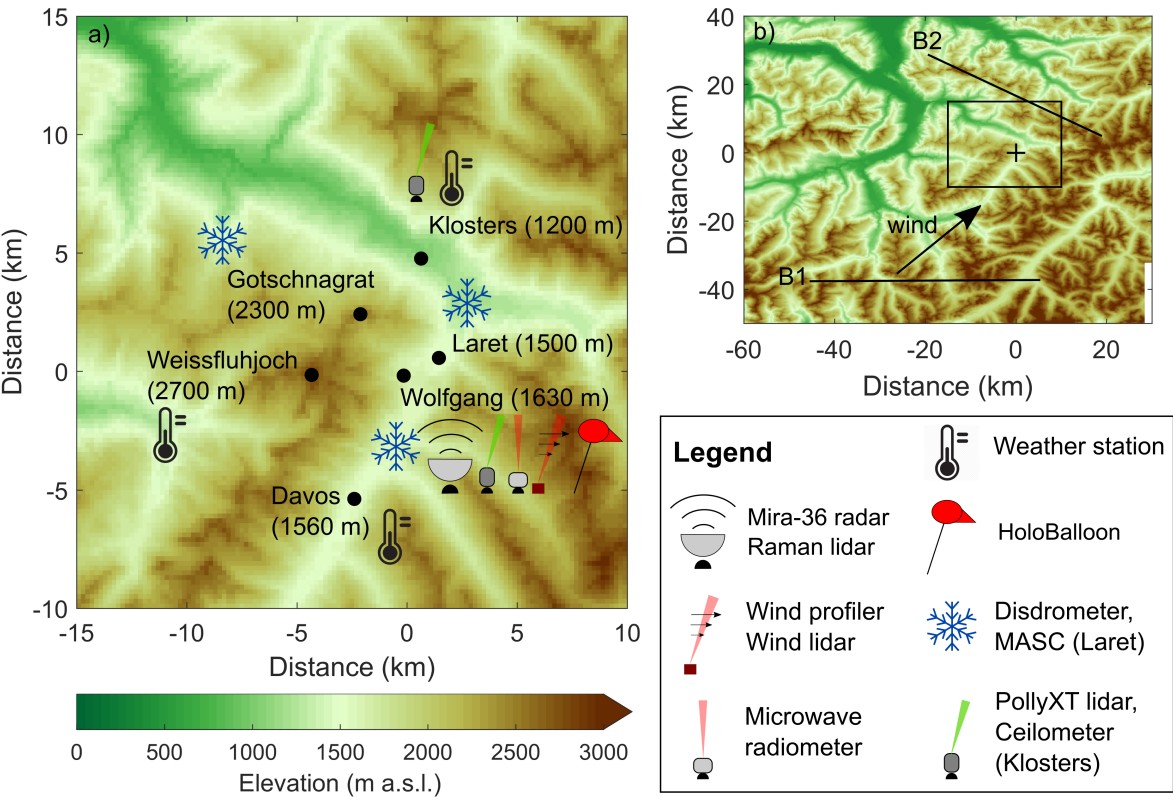

**Figure 1.** Overview of the measurement location and setup. The geographical location of Davos (black cross) and the surrounding topography is shown in (b). The large-scale wind direction during the event is shown by the black arrow and the relevant mountain barriers are indicated by B1 (upstream mountain barrier) and B2 (downstream mountain barrier). An enlarged section of the measurement sites (black rectangle in b) and the instrument setup are shown in panel (a). The elevation data was obtained from the digital height model DHM25 of the Federal Office of Topography swisstopo: https://shop.swisstopo.admin.ch/de/products/height_models/dhm25200, last access: 9 March 2020.

to improve regional precipitation forecast in complex terrain. For this purpose, a multi-dimensional set of instruments and measurements was deployed to provide a comprehensive dataset of orographic clouds.

## 2.1 Measurement location

A map of the instrument setup and the relevant measurement locations is shown in Figure 1. Davos is located in the Swiss Alps in the eastern part of Switzerland. The Alpine massif is oriented in a southwest-northeast direction and has a mean height of around 3000 m a.s.l. or 2400 m above the surrounding lowlands. The Alps represent a barrier for incoming weather systems, which predominantly approach the measurement location in Davos from northwest or south. During the presented case study, the weather system came from south-western direction and thus followed approximately the direction of the Davos valley.

Weather systems approaching Davos from the south are influenced by topography to a larger extent, because the main ridge of





the Alps is located south of Davos (indicated by B1 in Fig. 1b).

The main measurement locations consist of two mountain-top stations (Weissfluhjoch 2700 m, Gotschnagrat 2300 m), and three valley stations (Wolfgang 1630 m, Laret 1500 m, Klosters 1200 m), which are located within a distance of 10 kilometres (see Fig. 1a). The region around Klosters and Davos is characterized by complex topography. The Klosters valley is oriented

from northwest to southeast and the elevation gradually increases from the lowlands (500 m) to 1200 m. In contrast, the Davos valley is oriented in a northeast-southwest direction. The height rapidly increases from Klosters (1200 m) towards Wolfgang (1630 m), before it slowly decreases on the way to Davos (1560 m). In the following, we will briefly describe the relevant instruments, which have been used in the present study.

## 2.2 Instrument setup

A set of ground-based remote sensing and in situ instruments was installed at Wolfgang to study the microphysical cloud structure (Fig. 1a). A vertically-pointing Ka-band cloud radar Mira-36 (METEK GmbH, Germany, Melchionna et al., 2008; Görsdorf et al., 2015; Löhnert et al., 2015) provided vertical profiles of radar reflectivity factor, Doppler velocity, Doppler spectra, spectral width and linear depolarization ratio (LDR) with a vertical resolution of 31.17 m and a temporal resolution of 10 s. A PollyXT Raman and depolarization lidar (e.g., Engelmann et al., 2016) was deployed to study the aerosol and cloud

properties. Moreover, a 14-channel microwave radiometer (HATPRO, Radiometer Physics GmbH, Germany; Rose et al., 2005) provided information about the vertical temperature and humidity profiles as well as the column integrated water vapor content (IWV) and liquid water path (LWP). In situ observations of the low-level microphysical cloud structure were obtained with a tethered balloon system (HoloBalloon; Ramelli et al., 2020). The main component of the measurement platform is the HOLo-graphic cloud Imager for Microscopic Objects (HOLIMO), which can image cloud particles in the size range of 6 μm to 2 mm

(Henneberger et al., 2013; Beck et al., 2017; Ramelli et al., 2020). It provides information about the phase-resolved number concentration, water content, size distribution and particle shape. Additionally, a ceilometer (CL31, Vaisala, US) was installed at Klosters, which was used to identify the height of the cloud base (Fig. 1a).

Observations of the three-dimensional wind fields were obtained with a radar wind profiler (LAP-3000 Wind profiler, Vaisala, US) and a wind lidar (Windcube 100S, Leosphere, France) at Wolfgang (Fig. 1a). The wind profiler had a temporal resolution

of 5 min and a vertical resolution of 200 m, whereas the wind lidar provided wind measurements with a higher vertical resolution of 50 m. The wind lidar operated at an elevation angle of 75°. Additionally, Range Height Indicator scans (RHI) were performed every 30 minutes in four different azimuth directions (0°, 70°, 180° and 250°).

Precipitation was measured using a set of different ground-based precipitation instruments. Three Particle Size Velocity (Parsivel) disdrometers (OTT Parsivel2, OTT HydroMet, Germany; Tokay et al., 2014) were installed at Wolfgang (1630 m), Laret

(1500 m) and Gotschnagrat (2300 m), respectively (see Fig. 1a). Parsivel disdrometers can measure the size and the fall velocity of hydrometeors falling through the sample volume independently. The particle size is estimated from the signal attenuation, whereas the particle fall velocity is estimated from the duration of the measured signal. Precipitation particles in the size range between 0.2 mm and 25 mm can be measured by the disdrometer with a temporal resolution of 30 s. Additionally, a Multi-Angle Snowflake Camera (MASC) at Laret took photographs of hydrometeors from three different angles and measured their





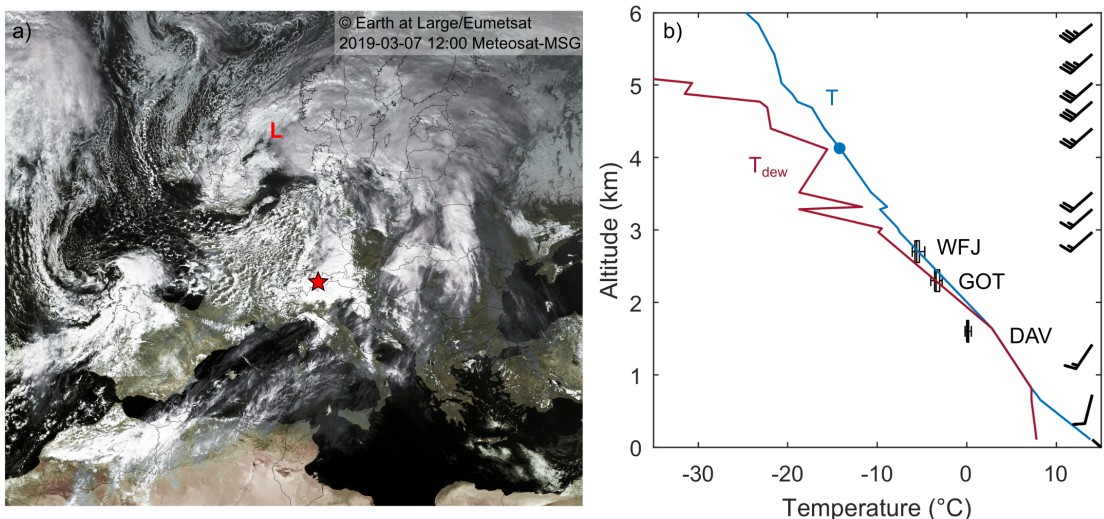

**Figure 2.** Overview of the synoptic weather situation on 7 March 2019 showing a satellite picture over Europe (a) and the temperature profile of a radiosonde ascent (b). The satellite picture was taken by the Meteosat at 12 UTC (Eumetsat). The radiosonde was launched from Milan (12 UTC; http://meteocentre.com/radiosonde/; last access: 16 March 2020) and shows the temperature (blue) and dew point temperature (red) profiles. The boxplots in (b) show the temperature measured at Davos (DAV, 1600 m), Gotschnagrat (GOT, 2300 m) and Weissfluhjoch (WFJ, 2700 m) during the measuement period. The cloud top temperature (-14 °C) and height (4000 m; estimated from cloud radar observations averaged between 17 UTC and 18.30 UTC) are indicated by the blue dot. The wind barbs are shown on the right side.

fall velocity simultaneously (Garrett et al., 2012). The MASC is sensitive to particles in the size range of 100 μm and 10 cm. Lastly, a snow drift station was installed at Gotschnagrat, which provided information about the snow redistribution at the ground and the low-level wind field (Walter et al., 2020).

## 3   Description of the case study

The weather situation on 7 March 2019 was characterized by a low-pressure system that moved from the North Sea towards
Scandinavia and brought a cold front towards Switzerland (Fig. 2a). Ahead of the cold front, a 15 hPa pressure gradient between the south and north side of the Alps produced a strong foehn event with wind gusts of up to 130 km h$^{-1}$. The cold front crossed Switzerland from the southwest in the morning and ended the pronounced foehn situation. Based on observations, rainfall of up to 50 mm was produced on the southern side of the Alps during the passage of the cold front (not shown). Moist southwesterly flow in the post-frontal air mass led to light showers on the south side of the Alps due to orographic lifting. Some spillover
precipitation was observed on the lee side of the Alps and reached the measurement locations in Davos. The presented case study was measured in the post-frontal air mass between 16 UTC and 20 UTC.

The temperature at Davos (1600 m) was around 0 °C during the entire observational period, whereas the temperature at Weissfluhjoch (2700 m) decreased from -4.5 °C to -6 °C between 16 UTC and 20 UTC. Due to the lack of a sounding in the Davos





area during the measurement period, the vertical temperature profile of a radiosonde ascent from Milan (Italy; at 12 UTC) is

shown instead (Fig. 2b). The sounding in Milan is assumed to be representative of the upper air situation in Davos, as the air flow was from the southwest. In addition, the temperatures measured at Gotschnagrat and Weissfluhjoch were in good agreement with the temperature profile of the radiosonde (within 1 - 2 K), whereas the temperature observed in Davos was slightly colder. A cloud top temperature of around -14 °C was estimated from the observed temperature profile.

An overview of the microphysical cloud structure is shown in Figures 3 and 4. The cloud radar observations indicate the

presence of a mid-level cloud with a cloud top at around 4000 m. The highest reflectivities (> 5 dBZ) were observed between 2500 m and 3500 m (Fig. 3a). The reflectivity decreased below 2500 m, indicating the presence of a sublimation layer. The Doppler velocity showed mainly regions with negative Doppler velocity (Fig. 3b). Positive Doppler velocities (i.e., updrafts) were only observed after 18 UTC near cloud top and after 19 UTC near the ground. Additionally, several regions of enhanced spectral width were observed (Fig. 3c). High values in the spectral width signal can be the result of enhanced turbulence and/or

indicate the presence of multiple particle populations with different fall speeds (e.g., Shupe et al., 2004; Shupe et al., 2006). The linear depolarization ratio (LDR) ranged between -32 dB and -22 dB (Fig. 3d) and provides information about the shape of the cloud particles. A perfectly spherical particle (e.g., small cloud droplet) has no depolarization and thus an LDR of -∞ dB, whereas a particle with a high aspect ratio has a LDR close to 0 dB. Furthermore, the LDR of a specific hydrometeor type depends on the elevation angle. Since a vertically-pointing cloud radar was used in this study, the observed LDR signal can

only differentiate between isometric particles (e.g., droplets, plates, dendrites) and prolate particles (e.g., needles, columns). The LDR signal was enhanced locally in some fallstreaks at altitudes below 3000 m, which is indicative of a change in the particle shape. The band of enhanced LDR at 2100 m, which was visible during the entire measurement period, shows the effects of ground clutter (i.e., echos received from objects on the ground or sidelobes reflected from nearby mountains).

The lidar signal was mainly attenuated due to the presence of a low-level liquid cloud (Fig. 4). When the low-level liquid cloud

dissipated (17:45 - 18:40 UTC), the lidar signal indicated the presence of an embedded liquid layer at an altitude of around 3500 m. This can be seen by the enhanced attenuated backscatter signal (Fig. 4a) and the low lidar depolarization ratio (Fig. 4b). The measured LWP was generally below 100 g m$^{-2}$ (Fig. 4c).

The case study was divided into three periods: Period P1_bl (16:00 UTC - 17:45 UTC) and period P3_bl (18:40 UTC - 20:00 UTC) were characterized by blocked low-level flow and the presence of a low-level liquid cloud at around 2000 m, whereas the low-

level blocking weakened and the low-level cloud dissipated during period P2_unbl (17:45 UTC - 18:40 UTC). In the following, we characterize the dynamics of the flow during the measurement period (Section 4.1). In a second step, we investigate the influence of shear-induced turbulence on the cloud microphysics and precipitation formation (Section 4.2). Lastly, the role of the low-level blocking for the formation of a low-level feeder cloud is discussed in Section 4.3.







**Figure 3.** Cloud radar observations of the radar reflectivity (a), Doppler velocity (b), spectral width (c) and linear depolarization ratio (d) measured on 7 March 2019. Note that the colorbar in (b) is centered at -1 m s$^{-1}$ to approximately account for the hydrometeor fall speed. The red line in panel (d) indicates the track of the 18 UTC LDR-fallstreak, which was investigated in Fig. 10. The measurement period is divided into three periods, where P1_bl and P3_bl indicate blocked low-level flow and P2_unbl indicates unblocked low-level flow (see Sect. 4.1 for more details).

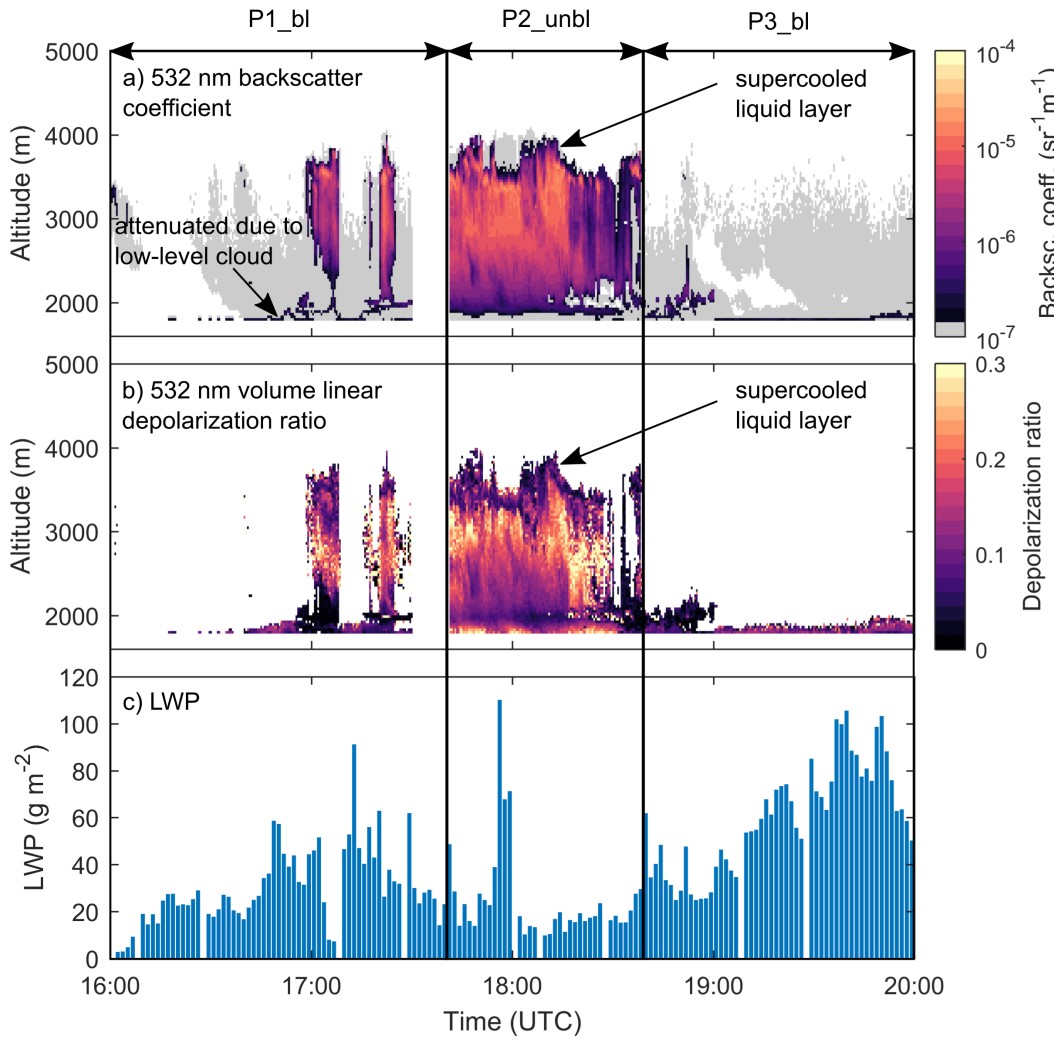

**Figure 4.** Observations of the lidar attenuated backscatter coefficient (a), the lidar linear depolarization ratio (b) and the liquid water path measured by the microwave radiometer (c). The measurement period is divided into three periods, where P1_bl and P3_bl indicate blocked low-level flow and P2_unbl indicates unblocked low-level flow (see Sect. 4.1 for more details).

# 4 Results

## 4.1 Low-level flow blocking triggering wind shear and turbulence

The horizontal wind fields were measured with a radar wind profiler and a wind lidar at Wolfgang (Fig. 5). Both wind profiler and wind lidar data are shown, as the wind lidar was attenuated during most of P1_bl and P3_bl due to the presence of a low-level liquid cloud (Section 4.3). The wind speed measured by the wind profiler increased from around $4\,\mathrm{m\,s^{-1}}$ at $2200\,\mathrm{m}$

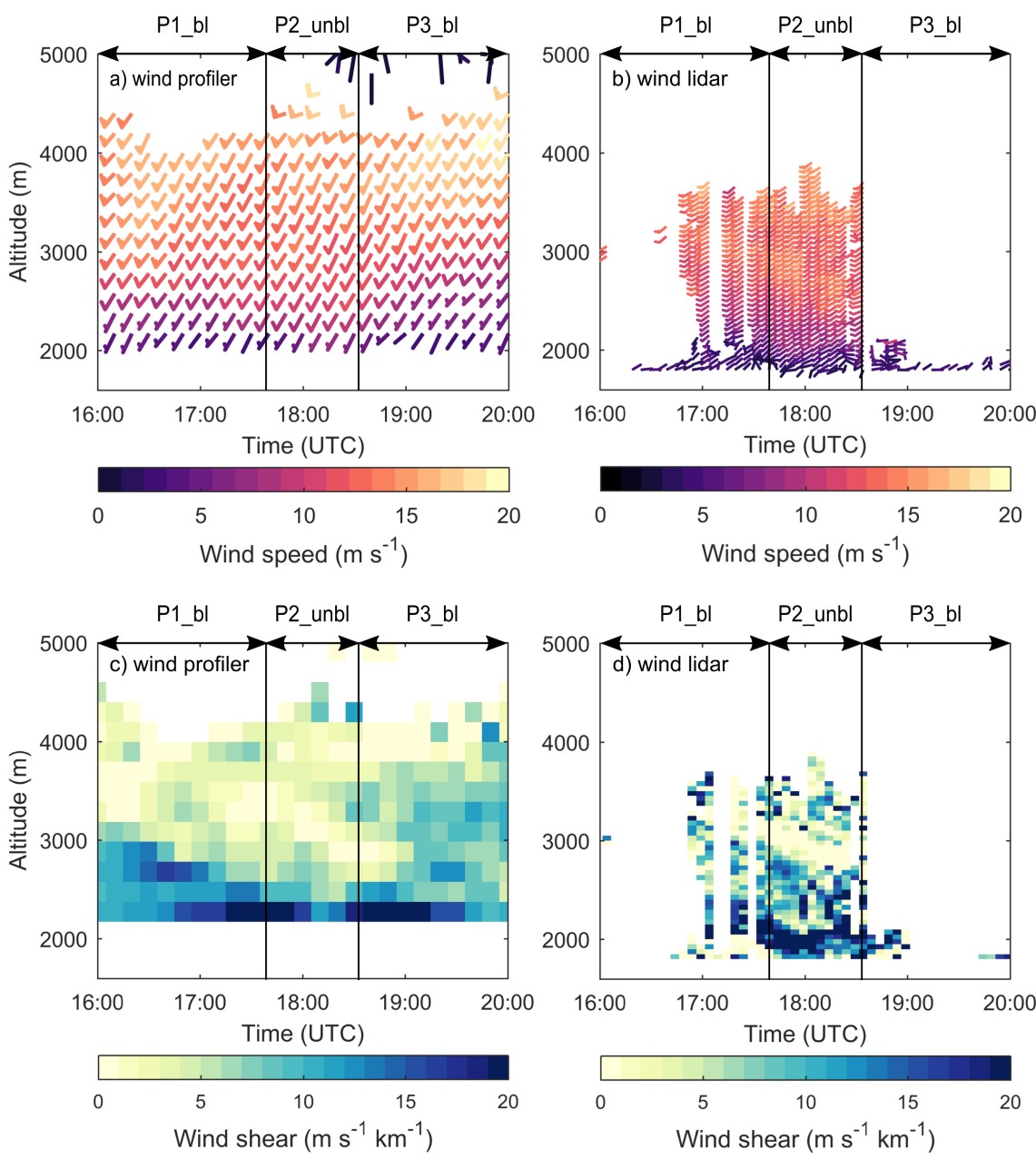

**Figure 5.** Vertical profiles of wind speed and direction (a, b) and wind shear (c, d) measured by the radar wind profiler (left) and wind lidar (right). The vertical wind shear was calculated from the wind observations, considering changes in the scalar wind speed between two adjacent height levels.





up to $18\,\mathrm{m\,s^{-1}}$ at 3500 m (Fig. 5a). The wind lidar revealed a second layer of increased wind speed between 2500 m and

3000 m (Fig. 5b), which was not captured by the wind profiler due to its lower resolution. The layer of increased wind speed

lowered between 17:00 UTC and 18:30 UTC, suggesting stronger cross-barrier flow. The large-scale wind direction was from

the southwest. Only in the lowest 100 m of the boundary layer, a flow from the northeast was observed by the wind lidar.

The counterflow at low levels with respect to the large-scale flow as well as the low wind speed close to the surface are indicative

of blocked low-level flow (e.g., Houze Jr and Medina, 2005). More specifically, the low-level flow at Wolfgang might have

been too weak to ascend over the mountain barrier located downstream of Wolfgang (B2 in Fig. 1b) and might have generated

a counterflow when it impinges on the mountain barrier. To test this hypothesis and to study the observed pattern in the low-

level wind field in more detail, wind measurements from different valley and mountain-top weather stations around Davos and

Klosters were analyzed (Fig. 6). Several valley stations (Davos, Davos Seehornwald, Wolfgang, Klosters Gatschiefer) observed

the presence of a counterflow (i.e., wind from northeastern direction), indicative of blocked low-level flow. The wind pattern at

Klosters Madrisa, which is located immediately below the mountain barrier B2, was more diverse, showing large variations in

the prevailing wind direction. The mountain-top stations (Weissfluhjoch, Gotschnagrat, Klosters Sant Jaggem) observed wind

from the south and southwestern direction in accordance with the large-scale wind direction. Therefore, the observed wind

pattern at the different locations support the hypothesis of a blocked low-level flow. It is important to note that a counterflow

was only observed during P1_bl and P3_bl at the valley stations in Davos (Davos, Davos Seehornwald, Wolfgang), indicating

that the low-level flow changed from a blocked (P1_bl, P3_bl) to an unblocked (P2_unbl) state during the event (see also Fig. 7

for the temporal evolution at Wolfgang). The flow at Klosters Gatschiefer was still blocked during P2_unbl, suggesting that the

blocking became weaker and moved closer to the mountain barrier located downstream.

From a theoretical perspective, the Froude number can be used to estimate whether a flow that encounters a mountain barrier

can pass over the mountain barrier or not (Smith, 1979; Durran, 1990; Rotunno and Houze, 2007; Houze Jr, 2012). The Froude

number $Fr$ is given by the following equation:

$$Fr = \frac{U}{hN} \tag{1}$$

where $U$ is the wind speed perpendicular to the mountain barrier, $h$ is the height of the mountain barrier and $N$ is the Brunt-

Väisälä frequency (Colle et al., 2013). The Froude number was below 1 for all periods (see Appendix A), indicating that the

low-level flow was blocked. The Froude number increased from 0.75 during P1_bl to 0.94 during P2_unbl (Fig. 7), suggesting

that the blocking became weaker during P2_unbl. We will refer to period P2_unbl as unblocked flow, as no counterflow was

observed during this period at Wolfgang (Fig. 7), even though it can be seen that the flow at low levels was still slowed down

and the Froude number was slightly below 1. Thus, period P2_unbl could also be regarded as a weaker blocking.

The wind profiler and wind lidar indicated the presence of a layer of strong vertical wind shear ($> 10\,\mathrm{m\,s^{-1}\,km^{-1}}$), which

separated the blocked layer from the stronger cross-barrier flow (Fig. 5c, d). The shear layer descended from 3000 m to 2000 m

between 16 UTC and 18 UTC (Fig. 5c) in accordance with the weaker blocking and stronger cross-barrier flow. As the cross-

barrier flow weakened and the strength of the low-level blocking increased again at around 18:30 UTC, the shear layer ascended

(Fig. 5c). Thus, the height of the shear layer is determined by a delicate balance of upstream (e.g., blocking) and downstream

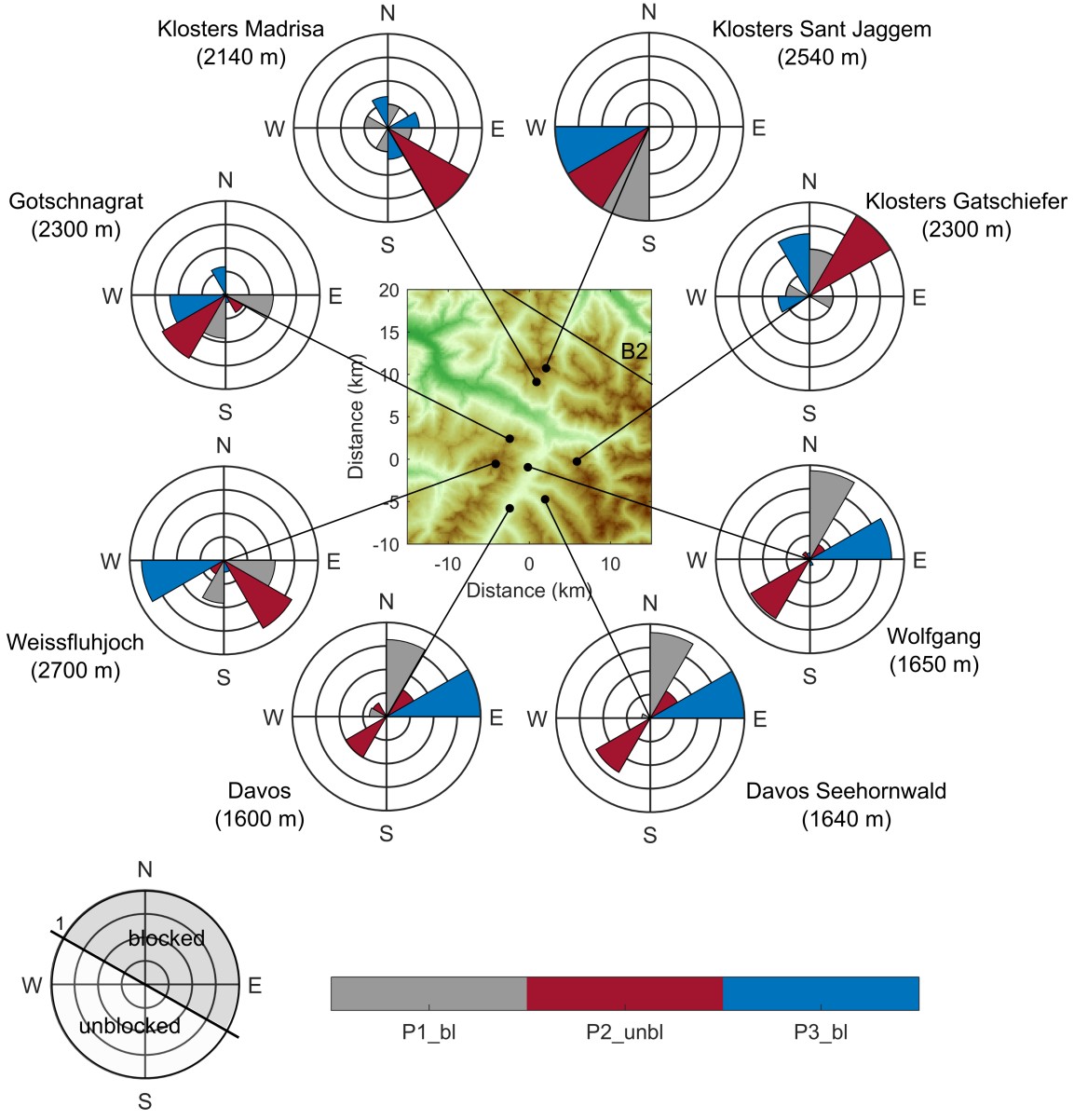

**Figure 6.** Wind observations at different weather stations around Davos and Klosters, indicating the wind direction observed during the time periods P1_bl (gray), P2_unbl (red) and P3_bl (blue). The wind directions in the different periods were normalized and binned in sectors of 90° (NE, SE, SW, NW). Each line indicates 25%. The example wind rose on the bottom left indicates the wind directions during a blocked (gray)/unblocked (white) low-level flow in the Davos valley. The wind observations at the weather stations in Davos had a temporal resolution of 10 min, whereas the stations around Klosters had a temporal resolution of 30 min. The measurements at Wolfgang were obtained from the wind lidar observations at an altitude of 1800 m and the measurements at Gotschnagrat from a 3D ultrasonic anemometer that was an integral part of the snow drift station.





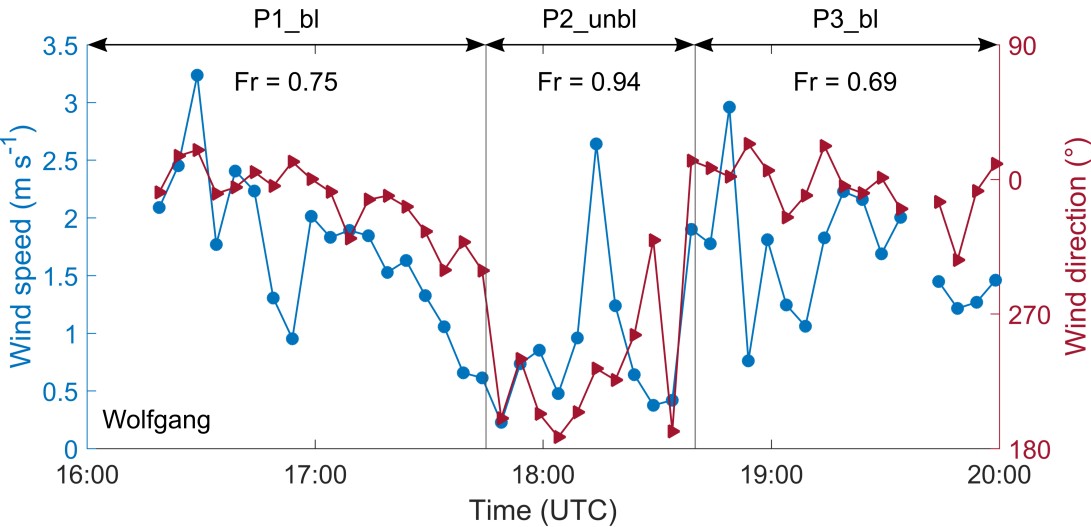

**Figure 7.** Temporal evolution of the wind speed (blue) and wind direction in azimuth degrees (red) at Wolfgang obtained from the wind lidar measurements at an altitude of 1800 m. The Froude numbers for the different periods are indicated at the top (see Appendix A for calculations).

(e.g., cross-barrier flow) effects, as Wolfgang is located in an inner-Alpine valley and surrounded by multiple mountain barriers. Regardless of the formation mechanism, wind shear can be a source of turbulence and can have important implications for cloud microphysics. Houze Jr and Medina (2005) defined a critical threshold of $10\,\mathrm{m\,s^{-1}\,km^{-1}}$ for the formation of shear-induced turbulent cells. The observed wind shear ($10\,\text{-}\,20\,\mathrm{m\,s^{-1}\,km^{-1}}$) was above this threshold value in the entire shear layer, suggesting that turbulent cells were embedded within the shear layer. In the following section, we will characterize the microphysical cloud structure and investigate the influence of shear-induced turbulence on the cloud microphysics and precipitation formation.

## 4.2 Influence of shear-induced turbulence on the cloud microphysics

Contour frequency by altitude diagrams (CFADs) are a useful tool for analyzing the magnitude and the vertical frequency distribution of cloud properties (e.g., Yuter and Houze Jr, 1995). CFADs of the cloud radar reflectivity (a, b), spectral width (c, d) and LDR (e, f) are shown in Figure 8 averaged over sub-periods of P1_bl (left) and P2_unbl (right). A strong increase in the radar reflectivity was observed near cloud top (Fig. 8a, b), indicative for ice formation and growth between 4000 m and 3500 m. Since the region of rapid increase in radar reflectivity was coincident with the height of the supercooled liquid layer measured by the Raman lidar ($L$ in Fig. 8a, b), it is likely that this supercooled liquid layer played an important role for ice nucleation and initial ice growth. The cloud particles continued growing below 3500 m until they reached a sublimation layer, which was identified by the layer of decreasing radar reflectivity and also used as definition for cloud base. The cloud base lowered between P1_bl and P2_unbl (Fig. 9), likely as a consequence of the stronger cross-barrier flow and the subsequent lowering

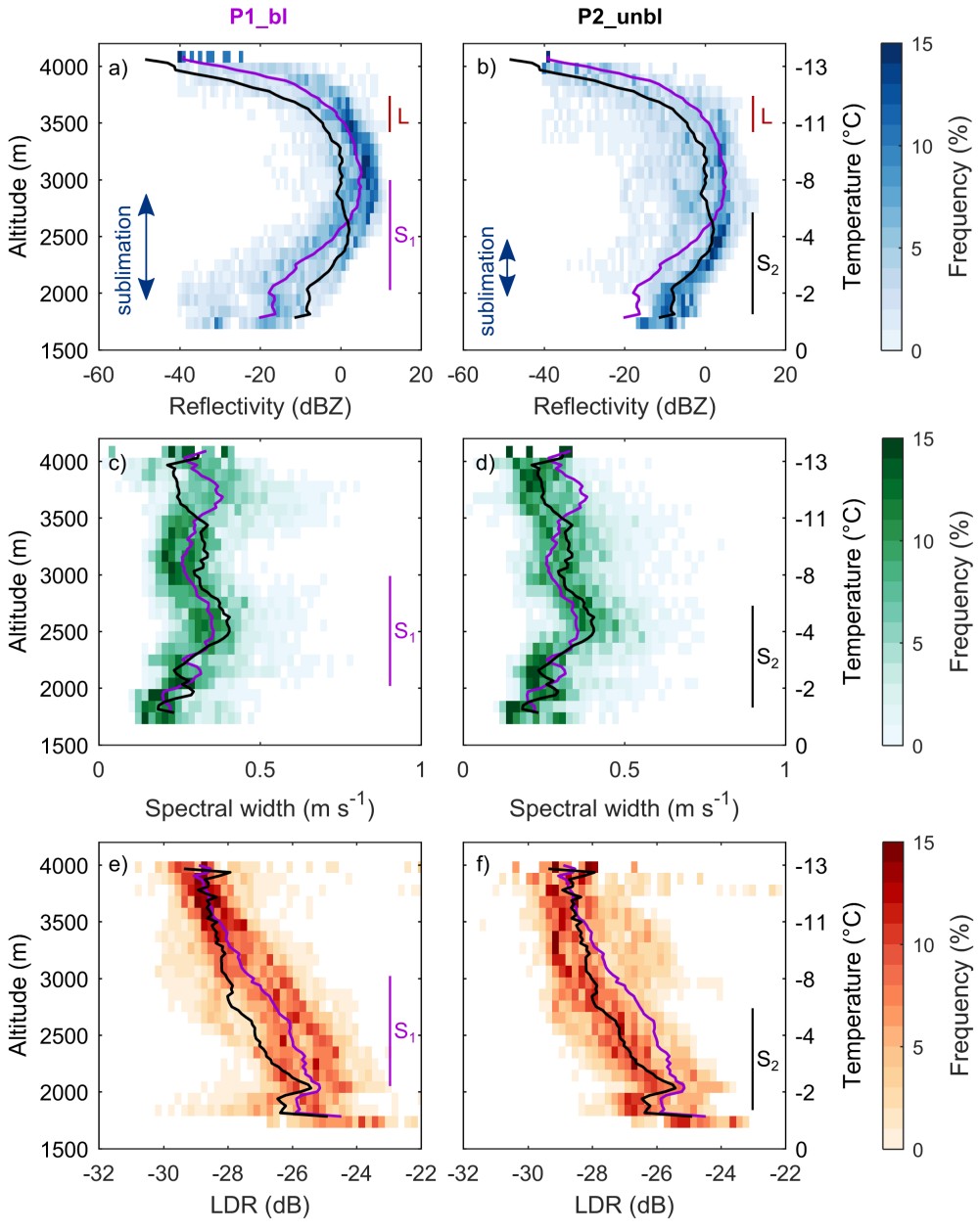

**Figure 8.** CFADs of the radar reflectivity (a, b), spectral width (c, d) and linear depolarization ratio (e, f) averaged over sub-periods of P1_bl (left, 17 UTC - 17:45 UTC) and P2_unbl (right, 17:45 UTC - 18:30 UTC). The following bin sizes were applied: (1) radar reflectivity from -40 dBZ to 20 dBZ in 1 dBZ intervals, (2) spectral width from 0 m s$^{-1}$ to 0.8 m s$^{-1}$ in 0.02 m s$^{-1}$ intervals and (3) LDR from -32 dB to -22 dB in 0.2 dB intervals. A height interval of 100 m was used for all radar properties. The solid lines represent the mean vertical profile of P1_bl (purple) and P2_unbl (black). The temperature profile measured by the radiosonde is shown on the right y-axis. The extent of the shear layer (S$_1$, S$_2$, from wind profiler), the supercooled liquid layer (L, from cloud lidar) and the sublimation layer (blue arrow, from cloud radar) are indicated.

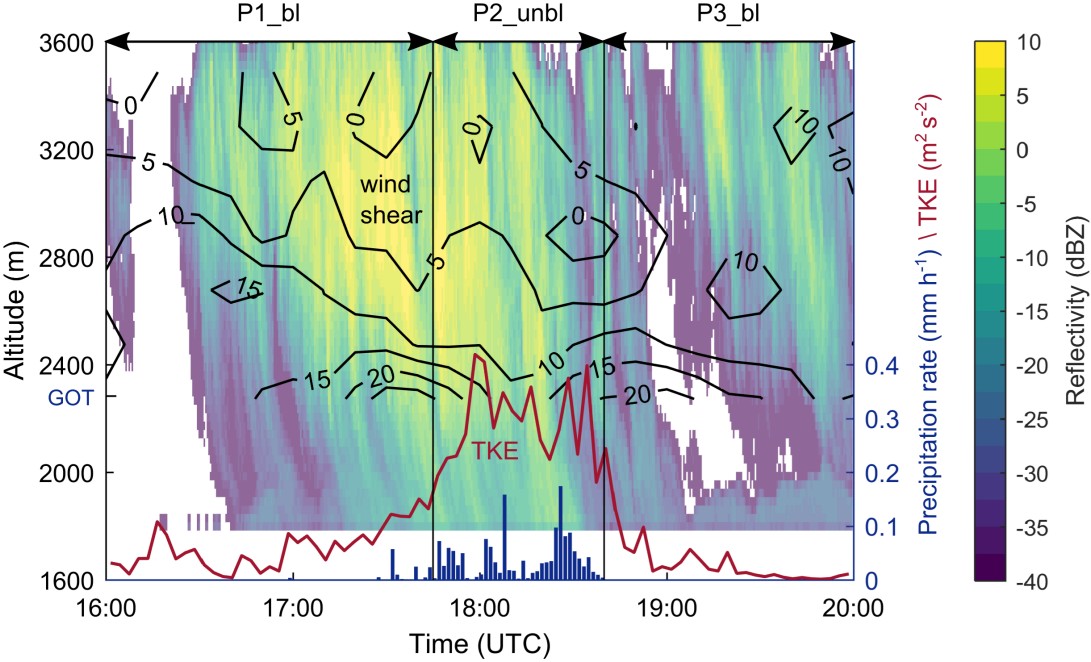

**Figure 9.** Temporal and spatial evolution of the vertical wind shear (in $\mathrm{m\,s^{-1}\,km^{-1}}$, black contour lines), the radar reflectivity (colorfill), the TKE (red line) and the precipitation rate (blue bars). The wind shear was calculated from the wind profiler observations (Fig. 5). The precipitation rate was measured with a disdrometer at Wolfgang (1630 m) and the TKE was measured with a 3D ultrasonic anemometer at Gotschnagrat (2300 m). Correlation coefficients between the different parameters (shear layer, radar reflectivity, precipitation) were calculated in Appendix B and indicated moderate to strong correlations significant at the 5 % level.

of the shear layer. Consequently, the extent of the sublimation layer decreased between P1_bl (800 m) and P2_unbl (400 m) (Fig. 8a, b), which enabled the hydrometeors to reach the surface prior to complete sublimation (Fig. 9). The highest radar reflectivities were observed within the upper part of the turbulent shear layer (marked with $S_1$ and $S_2$ in Fig. 8), suggesting that turbulence created updrafts high enough for exceeding ice saturation and thereby enhanced ice growth. The turbulent kinetic energy (TKE) measured by a 3D ultrasonic anemometer at Gotschnagrat (2300 m) increased from $0.1\,\mathrm{m^2\,s^{-2}}$ to $0.4\,\mathrm{m^2\,s^{-2}}$

between P1_bl and P2_unbl as the shear layer lowered (Fig. 9). The CFADs of the spectral width (Fig. 8c, d) show two local maxima embedded within the shear layer (at 2500 m and at 2000 m), indicating the presence of a broad hydrometeor size distribution within those regions, which can arise from enhanced turbulence. Additionally, an increase in the LDR was observed within the shear layer (Fig. 8e, f), which is indicative of a change in the hydrometeor shape. Thus, (1) the spatial coincidence of the shear layer and the maximum reflectivity (Fig. 9), (2) the temporal coincidence of the precipitation and the lowering

of the shear layer and cloud base (Fig. 9) and (3) the moderate to strong correlations between dynamics, microphysics and precipitation parameters (Appendix B) suggest that the processes active within the turbulent shear layer enhanced ice growth and precipitation formation.





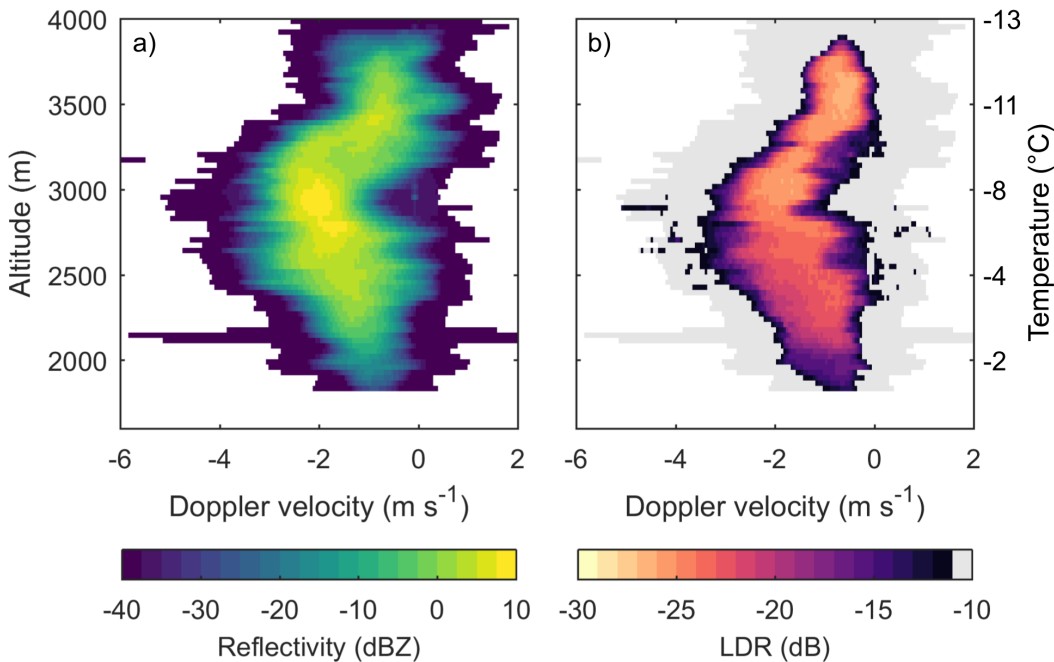

**Figure 10.** Vertical profile of the radar reflectivity (a) and LDR (b) Doppler spectra along the LDR-fallstreak at 18 UTC averaged over 1 min (see red line in Fig. 3d). The temperature profile measured by the radiosonde is shown on the right y-axis.

To further explore the microphysics within the turbulent shear layer, the Doppler spectra of the radar reflectivity and LDR along the 18 UTC fallstreak (Fig. 10; highlighted by red line in Fig. 3d) and the surface-based hydrometeor particles observations

(Fig. 11) were analyzed. As mentioned in Section 3, the LDR can provide information about the shape of cloud particles. A LDR of around -28 dB was observed near cloud top (Fig. 10b), which is characteristic for oblate particles such as dendrites and plates (e.g., Myagkov et al., 2016; Bühl et al., 2016). This is in agreement with the ice habit expected to form at a cloud top temperature of -14 °C (Magono and Lee, 1966; Bailey and Hallett, 2009) and with the ice particles observed by the MASC at the surface (Fig. 11a). Since a rather constant LDR was observed between 4000 m and 3000 m (Fig. 10b), we assume that

the ice crystals grew in size by vapor deposition without changing their habit. Below 3000 m, the LDR increased up to -20 dB within the fallstreak (Fig. 10b). Interestingly, the increase in the LDR was collocated with the region of maximum radar reflectivity (2900 m; Fig. 10a). The spatial coincidence between maximum radar reflectivity, shear layer and increase in LDR was also observed for other fallstreaks (Fig. 3d), suggesting that the turbulent shear layer did not only enhance ice growth but also contributed to a change in the cloud particle shape. An increase in the LDR can be explained by the presence of needles,

columns and/or irregular ice particles. Here we propose different mechanisms, which could have contributed to an increase in the observed LDR signal. Firstly, the temperature between 3000 m and 2500 m ranged from -8 °C to -4 °C and was thus in the temperature regime of column/needle growth. Thus, needles might have grown on the existing ice particles while they fell through that cloud layer. Indeed, the ice particles observed by the HoloBalloon platform near the ground show indications of




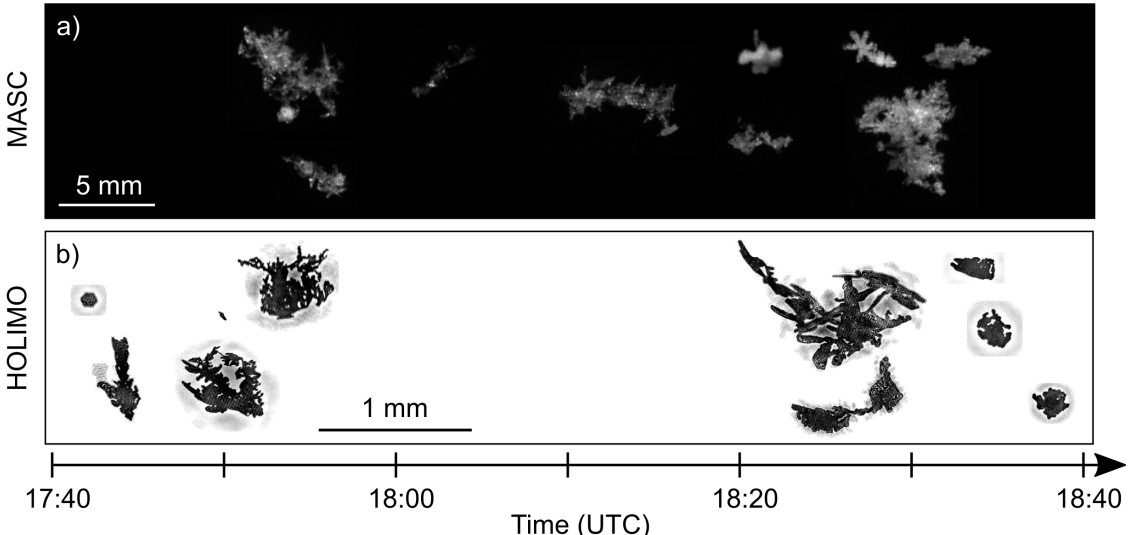

**Figure 11.** Photographs of ice crystals and snow particles, which have been taken with the MASC (a) and HOLIMO (b). No observations of HOLIMO were available between 17:50 UTC and 18:20 UTC. Please note the different size scale of the particles observed by the MASC and HOLIMO.

needle-like structures (e.g., at 17:50 UTC; Fig. 11b). If fragile ice crystals such as dendrites or needle-like structures collide
with large ice particles within the turbulent shear layer, small ice fragments might break off and lead to the production of
secondary ice particles upon collision (e.g., Vardiman, 1978; Yano et al., 2016). For example, the ice particles measured by
HOLIMO at 17:45 UTC could be a dendritic arm that broke off upon collision. Small secondary ice particles could then rapidly
grow by vapor deposition in the water-saturated environment into column-like particles, which are characterized by a higher
LDR. However, the analysis of the Doppler spectra showed no evidence of multiple spectral peaks (i.e., the presence of mul-
tiple particle populations with different fall speed), which would support the occurrence of secondary ice production. Lastly,
turbulence can also increase the generation of large ice particles (e.g., Pinsky and Khain, 1998), for example, when multiple ice
crystals collide and stick together (aggregation). Aggregation is most efficient at temperatures higher than -10 °C, because of
the higher sticking efficiency due to the presence of a thicker quasi-liquid layer at warmer temperatures (e.g., Lohmann et al.,
2016). Thus, if the aggregated particles that formed in the turbulent shear layer were irregular, this could explain the increased
LDR. Indeed, the MASC indicated the presence of large irregular aggregates between 17:40 UTC and 18:40 UTC (Fig. 11a).
For a more sophisticated analysis, a larger number of particles would be necessary, but due to the moderate precipitation rate at
Wolfgang and Laret, only a limited amount of ice and snow particles was observed. However, in general, the in situ and surface
observations of ice particles support the radar-based assumptions above, in that (1) dendrites formed near the cloud top and (2)
aggregation and needle growth occurred within the turbulent shear layer. It remains unclear whether mechanical break-up in
ice-ice collisions contributed to the formation of secondary ice particles.





### 4.3 Flow blocking as a driver for the formation of low-level feeder clouds

In the last part of this study, we focus on the lower part of the boundary layer and investigate the role of low-level blocking for the formation of a low-level feeder cloud. The low-level cloud structure was observed with the measurement platform HoloBalloon. Vertical profiles of the in situ cloud properties are shown in Figure 12. The cloud droplet number concentration (CDNC)

showed a rather inhomogeneous cloud structure during P1_bl. Cloud swaths with a CDNC of up to $100\,\mathrm{cm}^{-3}$ alternated with "cloud-free" regions with low CDNC (Fig. 12a). The mean cloud droplet diameter ranged between $10\,\mathrm{\mu m}$ and $17\,\mathrm{\mu m}$ (Fig. 12b) and was generally larger when the CDNC was low. No vertical profiles were performed between 17:50 UTC and 18:20 UTC, because the low-level cloud dissipated during this time period. This is also visible from the cloud lidar signal, which was not attenuated during P2_unbl in the lower part (see Fig. 12 and Fig. 4a). A second low-level cloud formed during P3_bl. This

cloud had a more homogeneous structure with a CDNC in the range between $60\,\mathrm{cm}^{-3}$ and $120\,\mathrm{cm}^{-3}$. The mean cloud droplet diameter steadily increased from $10\,\mathrm{\mu m}$ to $16\,\mathrm{\mu m}$. A low-level cloud was also observed by the cloud base observations of the ceilometer located in Klosters (not shown).

The interesting observation was that the low-level cloud dissipated during P2_unbl, when the low-level flow turned from a blocked to an unblocked state, pointing to the importance of the blocking situation in forming and sustaining the low-level

liquid cloud. We suggest that an overturning cell formed as a consequence of the low-level flow impinging on the mountain barrier located downstream of Wolfgang (as shown in Fig. 13a), which generated a low-level counterflow from Klosters towards Davos. Several weather stations near Davos confirm that a counterflow was present during P1_bl and P3_bl (Fig. 6). Since Wolfgang is located on a small-scale topographic feature (400 m), the low-level flow was forced to rise from Klosters (1200 m) to Wolfgang (1630 m) over the local topography and thereby acted as an updraft source, which led to the formation

of a low-level feeder cloud. Indeed, the cloud radar indicated the presence of updrafts below 2000 m (i.e., positive Doppler velocities) after 19 UTC (Fig. 3b). When the blocking weakened, the low-level cloud at Wolfgang dissipated because of the missing upward motion to sustain the production of liquid water.

This observation points to the importance of localized flow effects that interact with the topography in producing low-level feeder clouds over small-scale topographic features. Feeder clouds can enhance orographic precipitation through the seeder-

feeder mechanism (Bergeron, 1965; Bader and Roach, 1977; Hill et al., 1981), by providing an environment where hydrometeors that formed aloft (i.e., in the seeder region) can "feed" on the low-level liquid layer and enhance precipitation by riming and depositional growth. The seeder-feeder mechanism was for example visible in the cloud radar observations between 19 UTC and 20 UTC (Fig. 3a). In the present case study, this low-level feeder cloud did not play a role for precipitation enhancement, as a significant fraction of the hydrometeor mass sublimated before reaching the feeder cloud and precipitation was highest when

the flow was unblocked. However, we propose that local flow effects such as low-level blocking can also induce the formation of feeder clouds in other mountain valleys and in hilly regions and suggest that the extent of this effect depends on the strength of the blocking, the thermodynamics of the atmosphere and the altitude of the small-scale topographic feature that is located upstream of the mountain barrier.

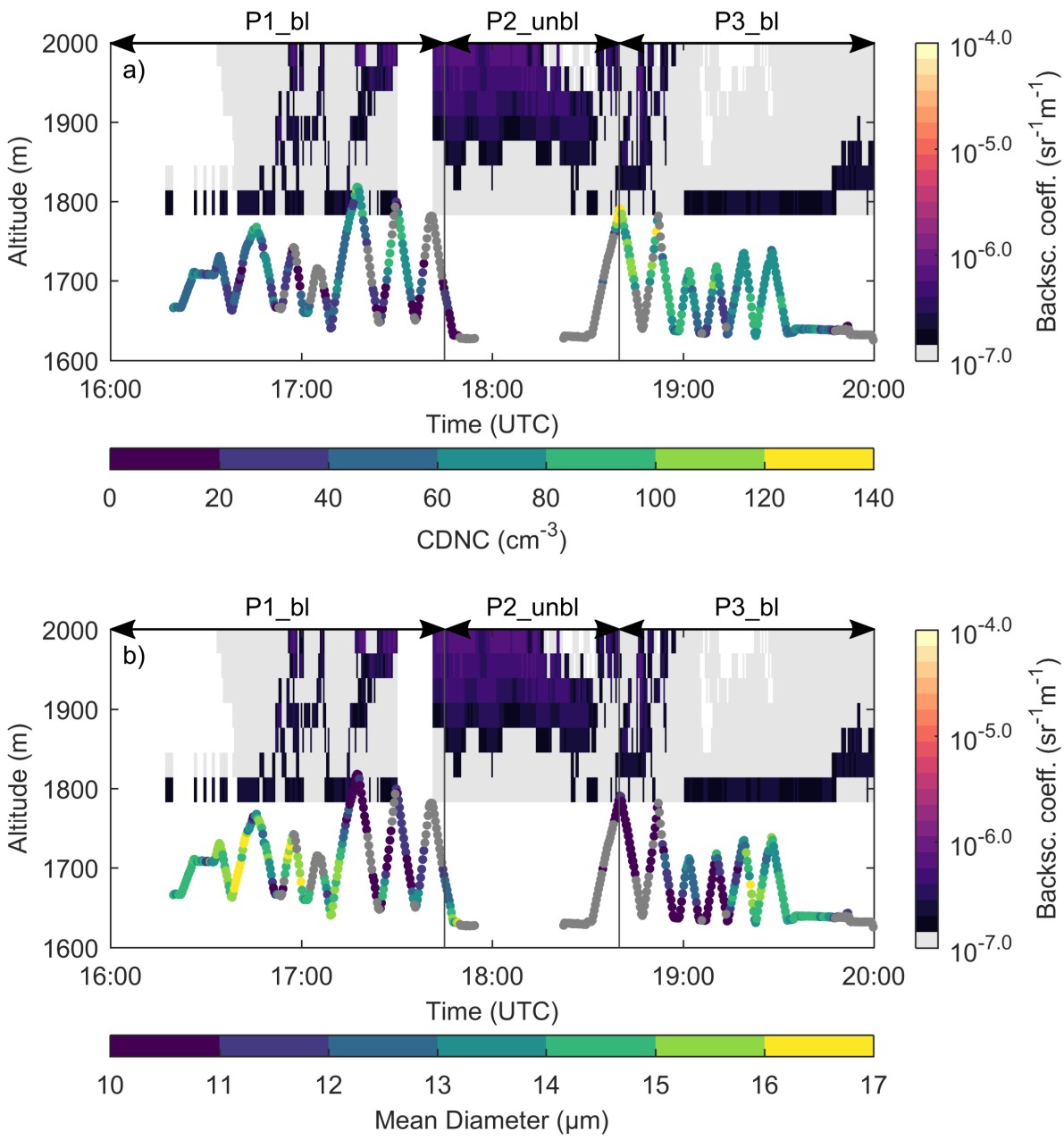

**Figure 12.** Vertical profiles of the CDNC (a) and mean droplet diameter (b) measured with the HoloBalloon platform. The lidar attenuated backscatter coefficient in the lowest levels is shown in the background. Data points with a liquid water content below $0.01\,\mathrm{g\,m^{-3}}$ are indicated by gray dots. No measurements were obtained between 17:50 UTC and 18:20 UTC.





## 5 Discussion

The microphysical evolution of the mixed-phase cloud in the inner-Alpine valley was determined by a complex interplay between orography, dynamics and microphysics. A conceptual overview of the observed cloud is shown in Figure 13 (a: blocked state; b: unblocked state). Primary ice nucleation was suggested to occur close to the cloud top in the embedded supercooled liquid cloud layer. Based on the cloud top temperature and cloud radar and hydrometeor observations, dendrites and hexagonal plates were assumed to form, which could rapidly grow by vapor deposition. The ice particles encountered a turbulent shear

layer while falling through the cloud, within which changes in the microphysical cloud properties were observed including enhanced radar reflectivity (i.e., increased ice growth) and LDR (i.e., change in particle shape). This suggests that the turbulent shear layer influenced the cloud microphysics.

Previous studies identified flow blocking and shear-induced turbulence as a microphysical pathway for enhancing snow growth and precipitation (e.g., Marwitz, 1983; Overland and Bond, 1995; Yu and Smull, 2000; Hogan et al., 2002; Neiman et al.,

2002; Neiman et al., 2004; Houze Jr and Medina, 2005; Loescher et al., 2006; Olson et al., 2007; Olson and Colle, 2009; Geerts et al., 2011; Medina and Houze Jr, 2015; Grazioli et al., 2015; Aikins et al., 2016). Different microphysical processes have been proposed to occur in the turbulent layer, such as enhanced growth by riming due to pockets of higher liquid water content (e.g., Houze Jr and Medina, 2005; Medina and Houze Jr, 2015 Grazioli et al., 2015) or enhanced growth by aggregation due to increased collisions between hydrometeors (e.g., Geerts et al., 2011; Aikins et al., 2016). Based on the cloud radar

observations and the ground-based snow particle measurements, we suggest that depositional growth and aggregation were the dominant ice growth mechanisms in the present case study. Riming was assumed to play only a minor role, due to the low LWP ($< 100 \, \text{g} \, \text{m}^{-2}$) observed by the microwave radiometer (Fig. 4c). Furthermore, the hydrometeors observed at the surface appeared primarily unrimed and indicated the presence of aggregates, dendrites and irregular ice particles (Fig. 11).

Furthermore, an increase in the LDR (>-25 dB; i.e., column-like ice particles) was observed within the shear layer (Fig. 10b

and Fig. 3d), which is indicative for a change in the hydrometeor shape. Based on the prevailing temperature near cloud base (from -8 °C to -5 °C) and the ice particle habits observed at the surface (Fig. 11), we suggest that needle growth occurred on the existing ice particles within the turbulent shear layer. Previous laboratory studies have observed needle growth at temperatures of -5 °C slightly below liquid water saturation (Knight, 2012). Thus, if fragile ice crystals such as dendrites or ice particles with needle-like structures collide with large ice particles, small ice fragments can break off and lead to the production of

secondary ice particles (e.g. Vardiman, 1978; Yano et al., 2016). Previous studies have observed a large number of small ice particles within turbulent shear layers, which were likely generated through secondary ice production mechanisms (e.g., Hogan et al., 2002; Grazioli et al., 2015). For example, Hogan et al. (2002) observed high concentrations of small ice particles ($100 \, \text{-} \, 1000 \, \text{L}^{-1}$) in and above a region of embedded convection, which were likely produced through the Hallett-Mossop mechanism during riming. Since the fall velocity of small ice particles is low, these secondary ice particles were found to

recirculate in the updrafts/turbulent region and to feed the regions above the shear layer with ice crystals (Hogan et al., 2002; Grazioli et al., 2015), where they could continue growing to precipitation-sized particles and act as a seed to trigger secondary ice production. The Hallett-Mossop process was likely not occurring in the present case study due to the lack of supercooled





**Figure 13.** Conceptual overview of the dynamical and microphysical processes observed in the cloud during a low-level blocked flow (a) and unblocked flow (b).





liquid water in the turbulent shear layer. Rather collisional ice multiplication of fragile ice crystals (i.e., needles, dendrites) may have been responsible for the increase in the LDR and in the radar reflectivity within the turbulent shear layer. Additionally,

a few laboratory-based studies suggested that ice fragmentation upon sublimation can lead to the production of secondary ice particles (e.g., Oraltay and Hallett, 1989; Dong et al., 1994; Bacon et al., 1998). However, secondary ice particles that formed within a subsaturated environment need to be transported to an ice supersaturated environment to influence the cloud microphysics (Korolev et al., 2020). In order to further investigate the role of secondary ice production in the turbulent shear layer, in situ observations of the cloud properties would be useful. Unfortunately, the tethered balloon system was limited to

lower altitudes and could not sample the cloud properties within the upper part of the shear layer.

While previous research has mainly focused on a single mountain barrier, here we studied the role of low-level blocking and shear-induced turbulence on the microphysics of a mixed-phase cloud in a more complex terrain with narrow valleys and a series of mountain barriers. We found that the height of the shear layer and the cloud base and as a consequence the amount of precipitation in the valley were determined by the strength of the cross-barrier flow and low-level blocking. Furthermore,

we found that local flow effects in mountain valleys (i.e., overturning cell due to blocked low-level flow) can induce the formation of low-level feeder clouds, which can enhance orographic precipitation through the seeder-feeder mechanism. Thus, this case study demonstrates that it can be challenging to study 'simple' conceptual mechanisms in complex terrain, due to numerous interactions between dynamics, microphysics and orography on different scales and the superposition of upstream and downstream effects. Nevertheless, it is important to perform field campaigns in complex terrain in order to improve our

understanding of these processes and of orographic precipitation. Field campaigns in complex terrain should be designed in such a way that an extensive set of complementary instruments are deployed over the measurement area. The present observations were obtained in a region of $10\,\mathrm{km} \times 10\,\mathrm{km}$ and mostly focused on the vertical structure. Information about the horizontal cloud structure (e.g., radar RHI scans, in situ aircraft observations) and observations over a larger area (e.g., covering the entire region between upstream and downstream mountain barriers) would be beneficial to obtain a more complete picture

of the cloud dynamics and microphysics.

## 6   Conclusions

In this paper, we studied the influence of low-level flow blocking and shear-induced turbulence on the microphysics of a mixed-phase cloud in an inner-Alpine valley. Observations from a multi-dimensional set of ground-based remote sensing, balloon-borne in situ and ground-based precipitation instruments were analyzed, which were acquired during the RACLETS

campaign in the Swiss Alps. The key findings are summarized as follows:

– The dynamical and microphysical structure of a mid-level cloud was characterized using ground-based remote sensing instrumentation (e.g., Ka-band polarimetric cloud radar, Raman lidar, radar wind profiler). The wind profiler observations indicated the transition from a blocked to an unblocked low-level flow during the observational period and the presence of a shear-induced turbulent layer, which separated the blocked layer in the valley from the stronger cross-barrier flow.

A supercooled liquid layer was embedded near cloud top, which provided a favorable environment for ice nucleation





and growth. Changes in the microphysical cloud properties were observed within the turbulent shear layer including enhanced LDRs (i.e., change in particle shape) and increased radar reflectivities (i.e., enhanced ice growth). Our results are consistent with previous studies that have observed enhanced ice growth and precipitation formation through riming and aggregation in turbulent layers. In addition, based on the enhanced LDR and the ice particle habits observed at the surface, we suggest that needle growth on existing ice particles occurred within the turbulent layer and that collisions of these fragile ice crystals (e.g., dendrites, needle-like structures) with large ice particles might have caused mechanical break-up and the subsequent production of small secondary ice particles. These ice fragments have the potential to recirculate in the shear layer and influence the cloud microphysics aloft. However, this process could not be directly measured in this study. Further studies are required to investigate the role of secondary ice production mechanisms in turbulent shear layers.

– The altitude of the shear layer was determined by a complex interplay of upstream and downstream effects. For example, the shear layer was observed to lower as stronger cross-barrier flow moved over the upstream mountain barrier and the low-level blocking weakened. The cloud base was found to be associated with the shear layer. Precipitation was only observed in the valley when the shear layer was at its lowest altitude. The resulting lower cloud base altitude reduced the time that the ice particles spent in the subsaturated environment, ultimately allowing for the precipitation to reach the surface. Thus, we propose that the amount of precipitation observed in a mountain valley is influenced by several factors such as (1) the strength of the cross-barrier flow and low-level blocking, (2) the vertical position of the turbulent shear layer and cloud base and (3) the thermodynamic state of the boundary layer.

– In situ instrumentation on a tethered balloon system observed a low-level feeder cloud, which dissipated when the low-level flow turned from a blocked to an unblocked state. We suggest that an overturning cell formed as a consequence of the low-level flow impinging on the downstream mountain barrier. As a small-scale topographic feature was located upstream of the mountain barrier, we suggest that the generated counterflow (i.e., blocked flow) was forced to rise over the local topography and thereby acted as an updraft source and as a driving force for the formation of a low-level feeder cloud. Although the feeder cloud did not enhance precipitation in the present case (due to the dry boundary layer aloft), we propose that local flow effect such as low-level blocking can induce the formation of feeder clouds in other mountain valleys or on the leeward slope of foothills upstream of the main mountain barrier, where they can enhance orographic precipitation through the seeder-feeder mechanism.

## Appendix A: Froude number

The dynamical response of a stable flow encountering a mountain barrier depends on the strength of the upstream airflow, the thermodynamic stability of the flow and the height of the mountain barrier. These components can be combined into a dimensionless number (Froude number; e.g., Colle et al., 2013):

$$Fr = \frac{U}{hN} \tag{A1}$$





**Table A1.** Parameters used to calculate the Froude number during P1_bl, P2_unbl and P3_bl.

| | Wolfgang | | Weissfluhjoch | | | | |
| | T (°C) | p (hPa) | T (°C) | p (hPa) | U (m s$^{-1}$) | N (s$^{-1}$) | Froude number |
|---|---|---|---|---|---|---|---|
| P1_bl | 0.1 | 827.7 | -5.2 | 725.6 | 7.6 | 0.013 | 0.75 |
| P2_unbl | -0.1 | 828.4 | -5.6 | 726.4 | 9.3 | 0.012 | 0.94 |
| P3_bl | -0.2 | 829.2 | -5.9 | 726.8 | 6.7 | 0.012 | 0.69 |

where $U$ is the wind speed perpendicular to the mountain barrier, $h$ is the height of the mountain barrier and $N$ is the Brunt-Väisälä frequency, which is a measure for the atmospheric stability. If the Froude number is large (Fr $\gg$ 1), the air flow can rise over the mountain barrier. When the Froude number is small (Fr $\ll$ 1), the upstream flow is blocked and cannot ascend over the mountain barrier. The parameters to calculate the Froude number for the periods P1_bl (16 - 17:45 UTC), P2_unbl (17:45 - 18:40 UTC) and P3_bl (18:40 - 20 UTC) are given in Table A1.

The effective terrain height $h$ between the valley and the mountain barrier (2700 m) was around 800 m (see B2 in Fig. 1b). The wind speed was obtained from the radar wind profiler and averaged over the 1600 - 2800 m height interval. The temperature and pressure were measured at Wolfgang (1630 m) and at Weissfluhjoch (2700 m) and a linear temperature gradient was assumed between the two measurement locations. The calculated Froude numbers were below 1 for all periods, indicative for a blocked low-level flow. The Froude number increased during P2_unbl (to 0.94), suggesting a weakening of the blocking.

## Appendix B: Correlation between dynamics, microphysics and precipitation

To quantify the interactions between dynamics, microphysics and precipitation, correlation coefficients were calculated between the different parameters of Figure 9 (see Table B1). A moderate positive Spearman's rank correlation ($\rho = 0.62$) was observed between the shear layer height and the altitude of maximum radar reflectivity. On the other hand, a negative correlation was found between the precipitation rate and the shear layer height ($\rho = -0.76$) and altitude of maximum radar reflectivity

**Table B1.** Correlation between (1) the height of the 10 m s$^{-1}$ km$^{-1}$ wind shear contour line, (2) the height of maximum radar reflectivity and (3) the precipitation rate measured at Wolfgang. The Spearman's rank correlation coefficient $\rho$ and their p-Values are shown. The correlation coefficients were calculated between 16:45 UTC and 18:30 UTC. A time lag of 10 min was applied to the precipitation measurements.

| | Spearman's rank correlation coefficient $\rho$ (p-Value) | | |
| | Height shear layer | Height max. reflectivity | Precipitation rate |
|---|---|---|---|
| Height shear layer | - | 0.62 ($<$ 0.001) | -0.76 ($<$ 0.001) |
| Height max. reflectivity | 0.62 ($<$ 0.001) | - | -0.55 ($<$ 0.001) |
| Precipitation rate | -0.76 ($<$ 0.001) | -0.55 ($<$ 0.001) | - |





($\rho = -0.55$). All correlations were significant at the 5% confidence level. The moderate to strong correlations between dynamics, microphysics and precipitation parameters suggest that interactions between dynamical and microphysical processes were

active within the turbulent shear layer, which enhanced ice growth and precipitation formation.

*Author contributions.*  FR analyzed the observational data and prepared the figures of the manuscript. FR, JH, JP, AL and JW performed the HoloBalloon measurements. JB and PS processed the remote sensing data and helped in interpreting the remote sensing observations. RE operated the OCEANET container during the RACLETS campaign. MH operated the radar wind profiler and processed the wind profiler data. FR, JH, RD and UL helped in analyzing and interpreting the data. FR prepared the manuscript with contributions from all authors.


*Acknowledgements.*  The authors would like to thank the participants of the RACLETS campaign for their technical support and many fruitful discussions. In particular, we are thankful to Michael Lehning (WSL/SLF, EPFL) and his whole team for their substantial support for realizing the RACLETS campaign by providing local contacts and support in requesting the necessary permissions. We would like to thank Paul Fopp

for providing his land for the RACLETS campaign. We would also like to thank Alexander Beck for helping with the organization of the field campaign. Moreover, the authors are thankful to Susanne Crewell (University of Cologne) and Bernhard Pospichal (University of Cologne) for their help in interpreting the microwave radiometer data. We would also like to acknowledge Benjamin Walter (SLF) for providing data of the snowdrift station located at Gotschnagrat. We thank the Swiss Federal Office of Meteorology and Climatology MeteoSwiss for providing the meteorological measurements, ceilometer data from Klosters, MASC observations and access to the COSMO1 and weather radar data.

Furthermore, we would also like to thank Eberhard Bodenschatz (MPI Goettingen) for his technical support during the development of the HoloBalloon platform. We would like to thank the Federal Office of Civil Aviation, particularly Judith Baumann and Jeroen Kroese, for their pragmatic approach in obtaining the flight permit. FR, JH, AL, JP, JW and UL acknowledge funding from the Swiss National Science Foundation (SNSF) grant number 200021_175824. RD would like to acknowledge funding from the European Research Council (ERC) through Grant StG 758005.



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
