# Peer review of "Influence of low-level blocking and turbulence on the microphysics of a mixed-phase cloud in an inner-Alpine valley"

_Atmospheric Chemistry and Physics, 2020_

## Referee Comment (RC1) · Dmitri Moisseev (Referee) · 9 Nov 2020

**Review of "Influence of low-level blocking and turbulence on the microphysics of a mixed-phase cloud in an inner-Alpine valley"**

By Ramelli et al

*General comments:*
The authors investigate the influence of low-level blocking and shear induced turbulence on the cloud microphysics. They find associated changes in the cloud microphysics associated. Additionally, a low-level feeder cloud is observed and seems to be caused by the low-level blocking.

This study is of great interest and importance for our understanding of the cloud processes in mountainous regions. I believe this paper can be published after major revisions. I am not sure if I completely agree with the authors' interpretation of radar observations and would like to see more explanations and answers to my questions. Please find my comments below.

*Specific comments:*

**Lines 157-162**. The interpretation of LDR (or any other dual-polarization radar variable) is a bit more complex. It is true that LDR depends on an apparent shape of a hydrometeor. However, to make matters more complex, it also depends on the particle refractive index, in case of ice particles the refractive index is related to the particle density. Therefore, increase in the ice particle density, i.e. by riming, may increase LDR. That is why changes in LDR are not necessary indicative of the changes in particle shape (see for example Fig. 2.8 page 65 in (Bringi and Chandrasekar, 2001)).

Another example is the change of LDR in the melting layer of precipitation, it is driven by the changes in the refractive index due to melting and not changes in particle shape.

Bringi, V. N., & Chandrasekar, V. (2001). *Polarimetric Doppler Weather Radar : Principles and Applications*. Cambridge University Press.

**Line 237**. "an increase in the LDR was observed within the shear layer (Fig. 8e, f), which is indicative of a change in the hydrometeor shape" Actually, to my eye the increase in LDR starts above the shear layer. It may be related to the increase in Doppler velocity as shown in Fig. 10. Such an increase in Doppler velocity may be indicative of riming (Mosimann, 1995; Kneifel and Moisseev, 2020), which affects LDR as well.

Mosimann, L., 1995: An improved method for determining the degree of snow crystal riming by vertical Doppler radar. Atmos. Res., 37, 305–323, https://doi.org/10.1016/0169-8095(94)00050-N.

Kneifel, S., and D. Moisseev, 2020: Long-Term Statistics of Riming in Nonconvective Clouds Derived from Ground-Based Doppler Cloud Radar Observations. J. Atmos. Sci., 77, 3495–3508, https://doi.org/10.1175/JAS-D-20-0007.1.

**Line 248-249**: "This is in agreement … with the ice particles observed by the MASC at the surface (Fig. 11a)."
Actually, MASC observations do show presence of rimed particles as well.

**Line 250**: "Below 3000 m, the LDR increased up to -20 dB within the fallstreak (Fig. 10b)."
It is not clear what you are referring to. Please indicate on the spectra which part has LDR values up to -20 dB.

If this is the faster falling part of the spectrum, then it cannot be due to newly formed ice crystals. In this case, in my opinion it would be riming. If the increase in LDR occurs in the slow falling part then it is cause by newly formed ice particles.

**Line 254:** "An increase in the LDR can be explained by the presence of needles, columns and/or irregular ice particles."
Do you observe this in Doppler spectra? You should see bimodal spectra, or at least some indication of bimodality.

**Line 259 – 261**: "If fragile ice crystals such as dendrites or needle-like structures collide 260 with large ice particles within the turbulent shear layer, small ice fragments might break off and lead to the production of secondary ice particles upon collision (e.g., Vardiman, 1978; Yano et al., 2016)."

Of course, it may also be H-M process if the LDR signatures caused by riming.

**Line 326-327**: "Riming was assumed to play only a minor role, due to the low LWP (< 100 g m−2 ) observed by the microwave radiometer (Fig. 4c)."

I am not sure if I agree that we can completely exclude riming based on LWP alone. Riming efficiency does not depend on LWP, but on liquid droplet diameter. So, given that droplets are big enough to take part in riming, riming could take place.
We can compute if riming would significantly affect particle properties, by computing rime mass fraction for the observed LWP values.
Let's assume that we a plate like crystal is formed at the cloud top. Just to have a first guess, we can choose P1a from Pruppacher and Klett.

TABLE 2.4a

Mass-size relationships for various types snow crystals. Data taken on Mt. Teine (1024 m, Hokkaido, Japan), based on data of Heymsfield & Kajikawa, 1987.

| Crystal type | Mass-size relation $m$(g), $d$(cm) | Diameter range (mm) |
| --- | --- | --- |
| C1h | $2.63 \times 10^{-2} \ d^{2.68}$ | 0.3-0.6 |
| P1a | $3.76 \times 10^{-2} \ d^{3.31}$ | 0.3-1.5 |

We can compute the expected FR as (see my simple Matlab code):

```
d = 0.3:0.01:1.5; %% mm
m = 3.76 * (10^-2) .* ((d/10).^3.31); %% grams for P1a

LWP = 50; % g/m^2
```

```
dm  = LWP.*(pi/4).* (d/100).^2;

%%% Rime mass fraction
FR = dm./(m+dm);
```

This computation gives rime mass fraction value close to 1. Of course, I have assumed the riming efficiency of 1, here. So, the actual FR value could be lower. But notice that I have used LWP of 50 g/m^2 and not 100 g/m^2.

If riming is taking place here, even for the observed low LWP values we can expect a significant impact on particle properties.

For more discussion on the effect of riming on snowflake properties, see:

Li, H., Moisseev, D., & von Lerber, A., 2018: How does riming affect dual- polarization radar observations and snowflake shape? J. Geophys. Res. Atmos., 123, 6070–6081. https://doi.org/10.1029/2017JD028186

Moisseev, D., A. von Lerber, and J. Tiira, 2017: Quantifying the effect of riming on snowfall using ground-based observations, J. Geophys. Res. Atmos., 122, doi:10.1002/2016JD026272.

---

## Referee Comment (RC2) · Anonymous Referee #2 · 16 Nov 2020

Review for ACP-2020-774

Influence of low-level blocking and turbulence on the microphysics of a mixed-phase cloud in an inner-Alpine valley

by

F. Ramelli, J. Henneberger, R. O. David, A. Lauber, J. T. Pasquier, J. Wieder, J. Buehl, P. Seifert, R. Engelmann, M. Hervo and U. Lohmann

General comments

This manuscript uses a wide variety of ground-based observations to investigate the

impact that orography can have on cloud microphysics in an Alpine environment. Understanding this impact is obviously important for increasing the accuracy of weather and climate forecasts in orographic regions, and the applications that depend on these forecasts.

According to the title, the major objective is to examine the effect of low-level blocking and turbulence on mixed-phase cloud microphysics, and a conceptual figure is given and discussed. The scope of the manuscript is rather broad, and tries to cover too many aspects without enough attention to detail. Many possible processes are described but, often, not enough evidence is presented in interpreting the observations. To be published, this manuscript reqires major revisions. In my opinion, the manuscript would benefit from a much tighter focus, and a discussion reduced to the relevant processes backed by evidence. A major issue is that low-level blocking and wind shear are not likely to be having an impact on the formation of the mixed-phase cloud (the supercooled liquid layer at cloud top) but possibly modifying the precipitation as it falls, i.e. through seeder-feeder processes.

Specific comment and questions

This case study observes the passage of a synoptic-scale frontal system, and some of the features described in the manuscript can be directly attributed to the large scale motion rather than the orography. The sloping shear feature above 2.5 km in Figure 5 is common to many synoptic scale frontal systems (e.g. Keyser and Shapiro, 1986), and similar wind and shear patterns are often seen in weather radar, radar windprofiler or scanning cloud radars in fronts passing over much flatter, homogeneous terrain. The vertical wind shear values are also similar to those observed in fronts over more homogeneous terrain (Chapman and Browning, 2001).

As shown in Figure 8, the highest radar reflectivity values are expected at the upper boundary of the sublimation zone, before the falling ice particles start to sublimate and reduce in size. Figure 8 and 9 show that the sloping shear feature appears to coincide

with this sublimation zone, with the location of the maximum radar reflectivity values lowering in altitude over time just above the 0.01 s-1 wind shear contour also lowering in time. This is what would be expected if the sloping shear feature indicates the frontal boundary between two air masses, one saturated, and one subsaturated. This correlation between the upper edge of the sloping shear zone and the maximum radar reflectivity values therefore suggests that the large scale forcing could be responsible in this case.

Hence, without additional observations, or using output from a high resolution numerical weather prediction model, it is difficult to determine whether the changes at low-level (blocked/unblocked flow) are responsible for any changes at upper levels.

Keyser, D., and M. A. Shapiro (1986), A review of the structure and dynamics of upper-level frontal zones. Mon. Wea. Rev., 114, 452–499, https://doi.org/10.1175/1520-0493(1986)114<0452:AROTSA>2.0.CO;2

Chapman, D. and Browning, K.A. (2001), Measurements of dissipation rate in frontal zones. Q.J.R. Meteorol. Soc., 127: 1939-1959. https://doi.org/10.1002/qj.49712757605

The wind shear values derived from the two instruments are not always consistent with each other. Is this due to the differences in spatial and temporal resolution, scan pattern or integration time? Please include the elevation angle that the wind profiler operates at and the scan pattern used by the Doppler lidar for deriving winds. The wind calculations assume a homogeneous wind field and it is known some scanning patterns are more susceptible to turbulence, which can mean that this assumption is no longer valid (Päschke et al., 2015). How much of an impact could the turbulent zones have on the horizontal wind and shear calculations? How about variations in the particle fall velocity?

Päschke, E., R. Leinweber, and V. Lehmann (2015), An assessment of the performance of a 1.5 $\mu$m Doppler lidar for operational vertical wind profiling based on a

1-year trial, Atmos. Meas. Tech., 8, 2251–2266, doi:10.5194/amt-8-2251-2015.

Section 4.2 attempts to describe the influence of shear on the particle microphysics, but insufficient evidence is given to support this. It is obviously difficult to use the Doppler velocity values directly, as these are compromised by the unknown vertical air motion, but the Doppler spectra do show important information. Figure 10 shows one example of the the Doppler spectra following one fall streak, and the broadening is consistent with changes in the particle microphysics; the broadening occurs in a temperature range that coincides with the temperature range for the Hallet-Mossop process for secondary ice production (-8 to -3 C). This increase in Doppler spectral width is clearly seen in Figure 3c between 3000 and 2500 m. However, this increase in Doppler spectral width is more or less constant in altitude throughout the entire time period, and not correlated with the wind shear, suggesting that temperature (possibly the Hallet-Mossop process) is responsible for this microphysical process, not shear.

Note that Doppler spectra wouldn't necessarily show discrete multiple peaks with secondary ice production in turbulent regions, or if sublimation is occuring (evaporation broadens the size distributions).

The conceptual picture shows ice above a supercooled liquid layer, which, although possible, is not that typical for mixed-phase clouds with relatively warm (above -27 C) cloud tops (e.g. Westbrook and Illingworth, 2011; Battaglia and Delanoë, 2013), and is not supported by the remote-sensing observations shown here. The one occasion during P2_unblocked where the base of the supercooled liquid layer is not at the top of the cloud layer seen by the cloud radar is when there is appreciable LWP. LWP of 100 gm-2 implies a liquid layer that is likely to be at least 400 m thick from theoretical adiabatic considerations (e.g. Merk et al., 2016, https://doi.org/10.5194/acp-16-933-2016), which would place the top of the liquid layer at the top of the cloud layer seen by the cloud radar. This means that the observed case study agrees with previous studies.

Battaglia, A., and J. Delanoë (2013): Synergies and complementarities of CloudSat-CALIPSO snow observations, J. Geophys. Res. Atmos., 118, 721–731, doi:10.1029/2012JD018092.

Westbrook, C. D., and A. J. Illingworth, (2011): Evidence that ice forms primarily in supercooled liquid clouds at temperatures > -27C, Geophys. Res. Lett., 38 (L14808), doi:10.1029/2011GL048021.

The data presented does indicate that low-level blocking influenced the presence of low-level cloud in the valley. The three periods selected showed clearly that low-level cloud was present during blocked low-level flow, but not once this blocking weakened.

One option would be to investigate the reasons for this further. The radar Doppler velocity plot suggests that the low-level liquid layer is being formed in updrafts, as almost all Doppler velocities appear to be slightly positive (i.e upwards) for this layer. Is this the case? Or is this due to the difficulty in reading the color scale? The typical vertical air velocity in this layer could be determined from either the Doppler spectra (similar to Fig. 10) or from CFADs of Doppler velocity (similar to Fig. 8). If the air motion is upwards, it would still be weak (< 1 m s-1), so would not necessarily counter the blocked flow argument but be a result of in-valley circulation. Does the LWP correspond to the updraft speed? How about the cloud droplet number or size (Figure 12)?

The wind direction changes and speed slows (in general) at Wolfgang during P2, which coincides with precipitation and no low-level liquid water (Figure 7). Is this just because there is enough time for the precipitation to fall before evaporating (shallower sub-saturated layer)? Is this precipitation solid or liquid? What is the size distribution?

Also of interest is why the seeder-feeder mechanism did not appear to operate in this particular case study, presumably due to the fact that the upper level precipitation rarely fell far enough to benefit. E.g Fig. 12 shows precipitation not quite reaching to 1800 m.

Figure 9. Is it likely that the TKE measured close to the surface at Gotschnagrat is

[Figure]

representative of the turbulence in the atmosphere? Isn't it more likely to be due to local shear close to surface? What are the wind speeds at this location? The direction is the same as at the surface, but Fig. 5a suggests high wind speeds at this height, is the increase in TKE just due to an increase in wind speed close to the surface? Why would this then correlate with the precipitation rate elsewhere in the valley? How does this relate to the conceptual figure?

I'm not convinced of the usefulness of any the correlation coefficients described here. How do they relate to any expected dynamical or microphysical processes? The shear layer appears to coincide with the sublimation zone. For this case study, any attempt to link the surface precipitation to the maximum reflectivity in the profile should at least take the varying sublimation depth into account.

Technical comments

Line 52: Is this wind shear value for wind shear in the horizontal or in the vertical?

Lines 91-92: Do you mean 'the mean ridge height'?

Line 116,118: Isn't Vaisala a Finnish company?

Figure 2 caption: This should state 'taken by the Meteosat 2nd Generation (MSG) satellite'.

Line 228: Cloud base? The ice cloud continues to the surface during P1 and P2. Since all ice is falling (precipitating), the ice cloud base is defined in terms of visibility, not in terms of relative humidity (changes in growth or evaporation rate). Hence it is only during P3 that there is an ice cloud base.

---

## Author Comment (AC1) · 23 Dec 2020

**Reviewer comments on 'Influence of low-level blocking and turbulence on the microphysics of a mixed-phase cloud in an inner-Alpine valley' by Fabiola Ramelli et al.**

Response to Dmitri Moisseev

We would like to thank Dmitri Moisseev for his constructive and helpful feedback and suggestions on the manuscript. We incorporated the suggestions within the revised manuscript, which significantly improved the quality of the manuscript. In the following, we will address the reviewer's comments and present our responses and changes in the revised manuscript. Reviewer comments are reproduced in blue and author responses are in black. All line numbers in the author's response refer to the revised manuscript.

**General comments**

1) *The authors investigate the influence of low-level blocking and shear induced turbulence on the cloud microphysics. They find associated changes in the cloud microphysics associated. Additionally, a low-level feeder cloud is observed and seems to be caused by the low-level blocking. This study is of great interest and importance for our understanding of the cloud processes in mountainous regions. I believe this paper can be published after major revisions. I am not sure if I completely agree with the authors' interpretation of radar observations and would like to see more explanations and answers to my questions. Please find my comments below.*

Thank you for your comments and for raising several points, which in particular improved the interpretation of the radar observations. We included riming and the Hallett-Mossop process as possible microphysical mechanisms active within the mid-level cloud. The interpretation and discussion of the case study changed accordingly, as addressed in the responses to the specific comments.

**Specific comments**

2) *Lines 157-162. The interpretation of LDR (or any other dual-polarization radar variable) is a bit more complex. It is true that LDR depends on an apparent shape of a hydrometeor. However, to make matters more complex, it also depends on the particle refractive index, in case of ice particles the refractive index is related to the particle density. Therefore, increase in the ice particle density, i.e. by riming, may increase LDR. That is why changes in LDR are not necessary indicative of the changes in particle shape (see for example Fig. 2.8 page 65 in (Bringi and Chandrasekar, 2001)).*

*Another example is the change of LDR in the melting layer of precipitation, it is driven by the changes in the refractive index due to melting and not changes in particle shape.*

Thank you for this comment and the references. Indeed, as you pointed out the LDR signal depends on both the shape and density of the hydrometeors. We have now added a short description about the dependence of the LDR on the particle refractive index and included the proposed references (page 7, line 166-169): *"Furthermore, the LDR depends on the particle refractive index. Liquid water has a higher refractive index (0.93) than ice (0.197) (Houze Jr, 2014). For ice particles, the refractive index is related to the particle density, such that hail and graupel have a higher refractive index compared to snowflakes (Bringi and Chandrasekar, 2001). Therefore, changes in the LDR are indicative for changes in the particle shape and/or particle density (e.g. riming)."*

3) *Line 237. "an increase in the LDR was observed within the shear layer (Fig. 8e, f), which is indicative of a change in the hydrometeor shape" Actually, to my eye the increase in LDR starts above the shear layer. It may be related to the increase in Doppler velocity as shown in Fig. 10. Such an increase in Doppler velocity may be indicative of riming (Mosimann, 1995; Kneifel and Moisseev, 2020), which affects LDR as well.*

Thank you for this comment and the references. We agree that the increase in the Doppler velocity might be related to riming. Especially, since the LDR of the faster falling part of the spectrum was observed to increase (Fig. 10). As no in situ observations were available within the upper part of the shear layer, we cannot draw any conclusions regarding the dominant microphysical processes. Thus, we discuss both riming and aggregation as possible ice growth mechanisms within the shear layer. We extended the interpretation as follows and included the suggested references (page 14, line 267-274):*" The increase in the Doppler velocity might be indicative of riming. Previous studies observed that an increase in the Doppler velocity can be indicative of riming, which leads to a higher terminal fall velocity of particles due to the rapid gain of ice particle mass (e.g., Mosimann, 1995; Kneifel and Moisseev, 2020). This is further supported by the increase in the LDR of the faster falling population of the spectrum as a consequence of the higher particle density. In addition, turbulence within the shear layer could increase the number of collisions between ice particles and promote the formation of aggregated particles (e.g., Pinsky and Khain, 1998)."*

4) *Line 248-249: "This is in agreement … with the ice particles observed by the MASC at the surface (Fig. 11a)." Actually, MASC observations do show presence of rimed particles as well.*

Thank you for pointing this out. We changed the sentence as follows (page 16, line 273-274): *"Indeed, the hydrometeors observed by the MASC and HOLIMO show indications of rimed particles and large aggregates (Fig. 11), suggesting that both processes were occurring."*

5) *Line 250: "Below 3000 m, the LDR increased up to -20 dB within the fallstreak (Fig. 10b)." It is not clear what you are referring to. Please indicate on the spectra which part has LDR values up to -20 dB.*

*If this is the faster falling part of the spectrum, then it cannot be due to newly formed ice crystals. In this case, in my opinion it would be riming. If the increase in LDR occurs in the slow falling part then it is cause by newly formed ice particles.*

Thank you for this comment. We specified in which part of the spectra the LDR increased (page 14, line 262-264): *"Below 3000 m, the LDR of the faster falling part of the spectrum increased up to -21 dB (Fig. 10b) and the spectrum broadened (Fig.10a)."* As you pointed out correctly, this suggests that the increase in the LDR was due to riming on the faster falling particles, which then led to the increase in the Doppler velocity. Accordingly, the interpretation of the LDR signal was changed (page 14, line 267-271): "*The increase in the Doppler velocity might be indicative of riming. Previous studies observed that an increase in the Doppler velocity can be indicative of riming, which leads to a higher terminal fall velocity of particles due to the rapid gain of ice particle mass (e.g., Mosimann, 1995; Kneifel and Moisseev, 2020). This is further supported by the increase in the LDR of the faster falling population of the spectrum as a consequence of the higher particle density.*"

6) *Line 254: "An increase in the LDR can be explained by the presence of needles, columns and/or irregular ice particles."*

*Do you observe this in Doppler spectra? You should see bimodal spectra, or at least some indication of bimodality.*

Thank you for this comment. The Doppler spectra shows no indications of bimodalities, which would support the formation of a newly formed particle population. However, possible bimodalities might be masked by turbulence or subsaturated regions. The particles observed by HOLIMO show indications of needle growth on existing ice particles. We changed the text as follows (page 16, line 283-286): *"The analysis of the Doppler spectra showed no evidence of discrete multiple spectral peaks (i.e., the presence of multiple particle populations with different fall speed), which would support the occurrence of secondary ice production. However, turbulent regions or sublimation could broaden the size distributions and thus mask the presence of discrete multiple peaks in the Doppler spectra."*

7) *Line 259 – 261: "If fragile ice crystals such as dendrites or needle-like structures collide with large ice particles within the turbulent shear layer, small ice fragments might break off and lead to the production of secondary ice particles upon collision (e.g., Vardiman, 1978; Yano et al., 2016)."*

   *Of course, it may also be H-M process if the LDR signatures caused by riming.*

   Thank you for this comment. Indeed, the temperature (-8 °C to -4 °C) was in the temperature range of the Hallett-Mossop process. Thus, it is possible that secondary ice particles were produced upon riming. We added a sentence describing the possible occurrence of the Hallett-Mossop process (page 16, line 275-277): *"The temperature between 3000m and 2500m ranged from -8 °C to -4 °C and was thus in the temperature regime of columnar growth and of the Hallett-Mossop process. Thus, secondary ice particles might be produced upon riming, which could then rapidly grow by vapor deposition into column-like particles."* We also state that due to the missing in situ observations within the upper part of the shear layer it was not possible to draw any conclusions regarding the occurrence of secondary ice mechanisms (page 17, line 290-292): *"It remains unclear whether the Hallett-Mossop process and mechanical break-up in ice-ice collisions contributed to the formation of secondary ice particles (see also Sect. 5)."*

8) *Line 326-327: "Riming was assumed to play only a minor role, due to the low LWP (< 100 g m−2 ) observed by the microwave radiometer (Fig. 4c)." I am not sure if I agree that we can completely exclude riming based on LWP alone. Riming efficiency does not depend on LWP, but on liquid droplet diameter. So, given that droplets are big enough to take part in riming, riming could take place. We can compute if riming would significantly affect particle properties, by computing rime mass fraction for the observed LWP values. Let's assume that we a plate like crystal is formed at the cloud top. Just to have a first guess, we can choose P1a from Pruppacher and Klett.*

Mass-size relationships for various types snow crystals. Data taken on Mt. Teine (1024 m, Hokkaido, Japan), based on data of Heymsfield & Kajikawa, 1987.

| Crystal type | Mass-size relation $m$(g), $d$(cm) | Diameter range (mm) |
|---|---|---|
| C1h | $2.63 \times 10^{-2} \ d^{2.68}$ | 0.3-0.6 |
| P1a | $3.76 \times 10^{-2} \ d^{3.31}$ | 0.3-1.5 |

We can compute the expected FR as (see my simple Matlab code):

```
d = 0.3:0.01:1.5; %% mm
m = 3.76 * (10^-2) .* ((d/10).^3.31); %% grams for P1a

LWP = 50; % g/m^2

dm  = LWP.*(pi/4).* (d/100).^2;

%%% Rime mass fraction
FR = dm./(m+dm);
```

*This computation gives rime mass fraction value close to 1. Of course, I have assumed the riming efficiency of 1, here. So, the actual FR value could be lower. But notice that I have used LWP of 50 g/m^2 and not 100 g/m^2.*

*If riming is taking place here, even for the observed low LWP values we can expect a significant impact on particle properties.*

Thank you for the comment and the calculation. The calculation clearly shows that riming cannot be excluded based on the LWP alone. Based on your previous comments (see also answers to comments #2 to #4), we included riming as a possible ice growth mechanism in the revised manuscript (especially due to the increase in the negative Doppler velocity and the increase in the LDR of the faster falling population of the spectrum). Furthermore, the ice particles observed by the MASC and HOLIMO show indications of riming. We also discuss the possible role of the Hallett-Mossop process in the revised manuscript (see answer to comment #7).

---

## Author Comment (AC2) · 23 Dec 2020

**Reviewer comments on 'Influence of low-level blocking and turbulence on the microphysics of a mixed-phase cloud in an inner-Alpine valley' by Fabiola Ramelli et al.**

Response to Reviewer #2

We would like to thank the anonymous referee for his/her constructive and helpful feedback and suggestions on the manuscript. We incorporated the suggestions within the revised manuscript, which significantly improved the quality of the manuscript. In the following, we will address the reviewer's comments and present our responses and changes in the revised manuscript. Reviewer comments are reproduced in blue and the author responses are in black. All line numbers in the author's response refer to the revised manuscript.

**General comments**

1) *This manuscript uses a wide variety of ground-based observations to investigate the impact that orography can have on cloud microphysics in an Alpine environment. Understanding this impact is obviously important for increasing the accuracy of weather and climate forecasts in orographic regions, and the applications that depend on these forecasts.*

   *According to the title, the major objective is to examine the effect of low-level blocking and turbulence on mixed-phase cloud microphysics, and a conceptual figure is given and discussed. The scope of the manuscript is rather broad, and tries to cover too many aspects without enough attention to detail. Many possible processes are described but, often, not enough evidence is presented in interpreting the observations. To be published, this manuscript requires major revisions. In my opinion, the manuscript would benefit from a much tighter focus, and a discussion reduced to the relevant processes backed by evidence. A major issue is that low-level blocking and wind shear are not likely to be having an impact on the formation of the mixed-phase cloud (the supercooled liquid layer at cloud top) but possibly modifying the precipitation as it falls, i.e. through seeder-feeder processes.*

   Thank you for your comment and for raising several points, which helped to make the manuscript clearer. In particular, we shortened Section 4.2 and tried to focus on the relevant processes. It is difficult to provide conclusive evidence, as no in situ observations were available within the mid-level cloud and thus the analysis was based on remote sensing and ground-based ice particle observations. Furthermore, we removed Appendix B (correlation calculations) from the revised manuscript. On the other hand, following your suggestion, we extended Section 4.3 and investigated the reason for the formation of the low-level feeder cloud by relating the updraft velocity to the in situ measurements of the cloud properties.

   We agree that the low-level blocking and wind shear were not having an impact on the formation of the mid-level mixed-phase cloud or the supercooled liquid layer at cloud top; but that the shear layer likely modified the falling hydrometeors (e.g., through depositional growth, riming or aggregation) by providing an ice supersaturated environment. We modified some sentences in the revised manuscript to make this point clearer (see responses to specific comments). Additionally, we clarified that the blocked low-level flow was only responsible for the formation of the low-level feeder cloud (see Sect. 4.3).

**Specific comments and questions**

2) *This case study observes the passage of a synoptic-scale frontal system, and some of the features described in the manuscript can be directly attributed to the large scale motion rather than the orography. The sloping shear feature above 2.5 km in Figure 5 is common to many synoptic scale frontal systems (e.g. Keyser and Shapiro, 1986), and similar wind and shear patterns are often seen in weather radar, radar windprofiler or scanning cloud radars in fronts passing over much flatter, homogeneous terrain. The vertical wind shear values are also similar to those observed in fronts over more homogeneous terrain (Chapman and Browning, 2001).*

Thank you for this comment and the references. As described in Section 3.1 the case study was measured in a post-frontal environment on 7 March 2019 between 16 UTC and 20 UTC. The cold front passed the measurement site in Davos at around 8 UTC (see figure of the radar reflectivity below). We included a map of reanalysis data at 700 hPa height in the revised manuscript (see Fig. 2a) to give a better overview of the synoptic situation and extended the discussion of the synoptic situation (page 6, line 138-145): *"The synoptic situation over Europe was dominated by an occluding low-pressure system (980 hPa) located east of the British Isles. As the low-pressure system continued to propagate towards Scandinavia, it drove a cold front over the Alps, which passed the measurement location at 8 UTC. Based on observations, rainfall of up to 50 mm was produced on the southern side of the Alps during the passage of the cold front (not shown). By 15 UTC, southwesterly flow in the post-frontal air mass continued to advect cold air and moisture into the Alpine region (see Fig. 2a), which produced light precipitation on the south side of the Alps with some spillover precipitation on the lee side (i.e. north side) of the Alps. The case study was measured in the post-frontal air mass between 16 UTC and 20 UTC, when some spillover precipitation reached the measurement locations in the Davos region."*

[Figure]

*Figure 1: Radar reflectivity on 7 March 2019. The red rectangle indicates the measurement period of the present case study.*

Since the synoptic frontal system passed the measurement location already in the morning, we assume that the sloping shear feature was related to the orography. This is further supported by the height of the shear layer, which was related to the altitude of the upstream mountain barrier. Since we cannot exclude an influence of the synoptic system, we extended the paragraph as follows (page 13, line 226-232): *"Sloping shear features have also been observed in connection with synoptic scale*

*frontal systems (e.g. Keyser and Shapiro, 1986; Chapman and Browning, 2001), where similar vertical wind shear values have been measured (Chapman and Browning, 2001). The presented observations cannot provide conclusive evidence about whether the observed wind and shear patterns were orographically or synoptically driven. We suggest that the sloping shear feature was influenced - at least to some extent - by the orography, as the height of the shear layer was related to the altitude of the upstream mountain barrier B1 and the strength of the blocking and downward propagating cross-barrier flow (Fig. 5c, d and Fig. 7)."* Furthermore, we changed the subtitle of Section 4.1 from "Low-level flow blocking triggering wind shear and turbulence" to "Low-level flow blocking and wind shear" in the revised manuscript in order to avoid a clear assignment of a cause.

3) *As shown in Figure 8, the highest radar reflectivity values are expected at the upper boundary of the sublimation zone, before the falling ice particles start to sublimate and reduce in size. Figure 8 and 9 show that the sloping shear feature appears to coincide with this sublimation zone, with the location of the maximum radar reflectivity values lowering in altitude over time just above the 0.01 s⁻¹ wind shear contour also lowering in time. This is what would be expected if the sloping shear feature indicates the frontal boundary between two air masses, one saturated, and one subsaturated. This correlation between the upper edge of the sloping shear zone and the maximum radar reflectivity values therefore suggests that the large scale forcing could be responsible in this case.*

   *Hence, without additional observations, or using output from a high resolution numerical weather prediction model, it is difficult to determine whether the changes at lowlevel (blocked/unblocked flow) are responsible for any changes at upper levels.*

   Thank you for this comment. We agree that the shear layer likely indicates the boundary between a saturated and a subsaturated air mass. However, we suggest that the flow separation (dry boundary layer in the valley/ moist cross-barrier flow aloft) occurs due to the upstream topography, since the synoptic scale-frontal system passed the measurement location already in the morning and the height of the shear layer was related to the altitude of the orography (see also response to previous review comment).

   The boundary between the two air masses was characterized by a turbulent shear layer. We suggest that the hydrometeors falling through the shear layer from above likely experience changes in the cloud properties as indicated by the enhanced radar reflectivity (increased ice growth) and the increase in LDR (change in particle shape or density). Thus, we agree that the low-level blocking did not directly influence the formation and microphysics of the mid-level cloud. Instead, the blocked layer influenced, in combination with the strength of the downward propagating cross-barrier flow, the altitude of the shear layer. We modified some sentences in the revised manuscript to make this point clearer (page 21, line 341-345): *"The ice particles encountered a turbulent shear layer while falling through the cloud, within which changes in the microphysical cloud properties were observed including enhanced radar reflectivity (i.e., increased ice growth) and LDR (i.e., change in particle shape or density). This suggests that the turbulent shear layer created an ice supersaturated environment and thereby influenced the cloud microphysics."*

4) *The wind shear values derived from the two instruments are not always consistent with each other. Is this due to the differences in spatial and temporal resolution, scan pattern or integration time? Please include the elevation angle that the wind profiler operates at and the scan pattern used by the Doppler*

*lidar for deriving winds. The wind calculations assume a homogeneous wind field and it is known some scanning patterns are more susceptible to turbulence, which can mean that this assumption is no longer valid (Päschke et al., 2015). How much of an impact could the turbulent zones have on the horizontal wind and shear calculations? How about variations in the particle fall velocity?*

*Päschke, E., R. Leinweber, and V. Lehmann (2015), An assessment of the performance of a 1.5 m Doppler lidar for operational vertical wind profiling based on a 1-year trial, Atmos. Meas. Tech., 8, 2251–2266, doi:10.5194/amt-8-2251-2015.*

Thank you for this comment. The differences in the vertical wind shear are mostly due to different vertical and temporal resolution of the wind profiler and wind lidar. The temporal and spatial resolution are specified on page 5, line 122-124: *"The wind profiler had a temporal resolution of 5 min and a vertical resolution of 200 m, whereas the wind lidar provided wind measurements with a higher vertical resolution of 50 m."* This pattern is also visible on Figure 5. Furthermore, we included the elevation angle and the scan pattern used by the wind lidar in the revised manuscript (page 5, line 124-126): *"The wind lidar operated in Doppler Beam Switching (DBS) mode with 4 beams at an elevation angle of 75° and a vertical beam. Additionally, Range Height Indicator scans (RHI) were performed every 30 minutes in four different azimuth directions (0°, 70°, 180° and 250°)."* The wind profiler had a vertical beam as specified on page 5, line 121 in the revised manuscript. DBS is a widely used technique to measure 3D wind properties. According to a manufacturer, both VAD and DBS can be used to retrieve wind speed. The scanning strategy will have a bigger impact on variance measurements (Newman et al., 2016; https://doi.org/10.5194/amt-9-1993-2016) but this is beyond the scope of this paper.

5) *Section 4.2 attempts to describe the influence of shear on the particle microphysics, but insufficient evidence is given to support this. It is obviously difficult to use the Doppler velocity values directly, as these are compromised by the unknown vertical air motion, but the Doppler spectra do show important information. Figure 10 shows one example of the Doppler spectra following one fall streak, and the broadening is consistent with changes in the particle microphysics; the broadening occurs in a temperature range that coincides with the temperature range for the Hallett-Mossop process for secondary ice production (-8 to -3 C). This increase in Doppler spectral width is clearly seen in Figure 3c between 3000 and 2500 m. However, this increase in Doppler spectral width is more or less constant in altitude throughout the entire time period, and not correlated with the wind shear, suggesting that temperature (possibly the Hallett-Mossop process) is responsible for this microphysical process, not shear.*

Thank you for this comment. As shown in Fig. 8 and Fig. 9, the maximum radar reflectivity was observed to coincide with the upper part of the shear layer. This suggests that the shear layer created an ice supersaturated environment, which enabled the falling hydrometeors to grow to larger sizes. As you pointed out correctly, the Doppler spectra (Fig. 10) contains important information regarding the microphysics. The broadening of the Doppler spectrum and the increase in the Doppler velocity (and thus in the particle fallspeed) at 3000 m suggests that riming occurred within this layer as presented in the revised manuscript (page 14, line 264-271): *"Interestingly, the increase in the LDR was collocated with the region of maximum radar reflectivity (2900 m; Fig. 10a), of maximum (negative) Doppler velocity (2900m; Fig. 10a) and the upper part of the shear layer. The spatial coincidence between maximum radar reflectivity, shear layer and increase in LDR was also observed for other fallstreaks (Fig. 3d), suggesting that the shear layer created an ice supersaturated*

*environment, within which the hydrometeors grew to larger sizes. The increase in the Doppler velocity might be indicative of riming. Previous studies observed that an increase in the Doppler velocity can be indicative of riming, which leads to a higher terminal fall velocity of particles due to the rapid gain of ice particle mass (e.g., Mosimann, 1995; Kneifel and Moisseev, 2020). This is further supported by the increase in the LDR of the faster falling population of the spectrum as a consequence of the higher particle density.”*

Indeed, the temperature was in the range of the Hallett-Mossop process, so to account for this we now mention on page 16, line 275-277 that the Hallett-Mossop process was potentially occurring: *“The temperature between 3000 m and 2500 m ranged from -8 °C to -4 °C and was thus in the temperature regime of columnar growth and of the Hallett-Mossop process. Thus, secondary ice particles might be produced upon riming, which could then rapidly grow by vapor deposition into column-like particles.”*

6) *Note that Doppler spectra wouldn't necessarily show discrete multiple peaks with secondary ice production in turbulent regions, or if sublimation is occurring (evaporation broadens the size distributions).*

   Thank you for pointing this out. We included a sentence describing this effect in the revised manuscript (page 16, line 283-286): *“The analysis of the Doppler spectra showed no evidence of discrete multiple spectral peaks (i.e., the presence of multiple particle populations with different fall speed), which would support the occurrence of secondary ice production. However, turbulent regions or sublimation could broaden the size distributions and thus mask the presence of discrete multiple peaks in the Doppler spectra.”*

7) *The conceptual picture shows ice above a supercooled liquid layer, which, although possible, is not that typical for mixed-phase clouds with relatively warm (above -27 C) cloud tops (e.g. Westbrook and Illingworth, 2011; Battaglia and Delanoë, 2013), and is not supported by the remote-sensing observations shown here. The one occasion during P2_unblocked where the base of the supercooled liquid layer is not at the top of the cloud layer seen by the cloud radar is when there is appreciable LWP. LWP of 100 gm-2 implies a liquid layer that is likely to be at least 400 m thick from theoretical adiabatic considerations (e.g. Merk et al., 2016, https://doi.org/10.5194/acp-16-933-2016), which would place the top of the liquid layer at the top of the cloud layer seen by the cloud radar. This means that the observed case study agrees with previous studies.*

   Thank you for pointing this out. Indeed, this was a technical mistake in the drawing on our part. We changed the schematic in the revised manuscript, so that the supercooled liquid layer extends up to the cloud top.

8) *The data presented does indicate that low-level blocking influenced the presence of low-level cloud in the valley. The three periods selected showed clearly that low-level cloud was present during blocked low-level flow, but not once this blocking weakened.*

   *One option would be to investigate the reasons for this further. The radar Doppler velocity plot suggests that the low-level liquid layer is being formed in updrafts, as almost all Doppler velocities appear to be slightly positive (i.e upwards) for this layer. Is this the case? Or is this due to the difficulty in reading the color scale? The typical vertical air velocity in this layer could be determined from either the Doppler spectra (similar to Fig. 10) or from CFADs of Doppler velocity (similar to Fig. 8). If the air motion is upwards, it would still be weak (< 1 m s-1), so would not necessarily counter the blocked flow argument*

*but be a result of in-valley circulation. Does the LWP correspond to the updraft speed? How about the cloud droplet number or size (Figure 12)?*

Thank you for this comment and your suggestions. We extended Section 4.3 in the revised manuscript and investigated the reason for this transition further. Indeed, the updraft velocity seems to play a crucial role for the formation of the feeder cloud (estimated from the maximum Doppler velocity of the spectrum). Higher updrafts were observed during P1_bl and P3_bl, when a low-level feeder cloud was present (page 18, line 310-313): *"Indeed, the cloud radar indicated the presence of higher Doppler velocities and thus higher updraft velocities during P1_bl (0.5 m s$^{-1}$) and P3_bl (0.7 m s$^{-1}$) (Fig. 12). When the blocking weakened and the updraft velocity decreased during P2_unbl (0.37 m s$^{-1}$), the low-level cloud at Wolfgang dissipated likely due to insufficient upward motion to sustain the production of liquid water."* Furthermore, we calculated correlations between the maximum Doppler velocity and the microphysical properties in the revised manuscript (see Fig. 13 and page 18, line 313-322): *"The correlation plots between different dynamical (mean and maximum Doppler velocity) and microphysical properties (LWC, CDNC, mean diameter) in Figure 13 further support the assumption that the updrafts driven by the in-valley circulation induced the formation of the low-level liquid cloud. Moderate positive Spearman's rank correlation coefficients were observed between the maximum Doppler velocity (vmax) and the LWC(0.42) and CDNC (0.46), whereas a weak correlation was found between the maximum Doppler velocity and the mean diameter D (0.17). Similar correlation coefficients were observed between the mean Doppler velocity and the microphysical properties (not shown). The increase in CDNC with increasing updraft velocity (Fig. 13c) suggests that droplet formation was limited by the vertical velocity that generates supersaturation, i.e. velocity-limited conditions were encountered at Wolfgang. This aspect is dealt with in more detail in a paper by Georgakaki et al. (2020), where they investigated the drivers of droplet formation in mixed-phase clouds using observations from the RACLETS campaign."*

The findings suggest that local updrafts were produced by the in-valley circulation during blocked low-level flow, which were the drivers for the formation of the low-level feeder cloud.

9) *The wind direction changes and speed slows (in general) at Wolfgang during P2, which coincides with precipitation and no low-level liquid water (Figure 7). Is this just because there is enough time for the precipitation to fall before evaporating (shallower subsaturated layer)? Is this precipitation solid or liquid? What is the size distribution?*

As mentioned in Section 4.2 (page 14, line 246-249), this is likely a consequence of the lower extent of the subsaturated layer (maximum radar reflectivity moves to lower altitudes, see Fig. 9), which enabled the hydrometeors to reach the surface prior to complete sublimation during P2_unbl. The amount of precipitation recorded by the disdrometer at Wolfgang was generally low (< 0.2 mm/h). The precipitation reaching Wolfgang was solid in accordance with the temperature slightly below 0 °C (see also ice particles detected by MASC and HOLIMO in Fig. 10). A plot of the particle size distribution of hydrometeors measured by the disdrometer at Wolfgang is shown below in Fig. 2. We included the following sentence in the revised manuscript (caption Fig. 11): *"The particles measured by the disdrometer at Wolfgang were primarily in the size range between 0.5 mm and 1.5 mm (not shown)."*

[Figure]

*Figure 2: Particle size distribution of hydrometeors measured by the disdrometer at Wolfgang (averaged between 17:45 UTC and 18:40 UTC).*

10) *Also of interest is why the seeder-feeder mechanism did not appear to operate in this particular case study, presumably due to the fact that the upper level precipitation rarely fell far enough to benefit. E.g Fig. 12 shows precipitation not quite reaching to 1800 m.*

We assume that the seeder-feeder mechanism only played a minor role for precipitation enhancement in the present case study, as the lower part of the boundary layer was characterized by subsaturated conditions and thus a significant fraction of the hydrometeor mass sublimated before reaching the feeder cloud. However, in the revised manuscript, we included a sentence stating that this effect was important in other cases of the RACLETS campaign (page 20, line 328-332): *"We assume that in the present case study the low-level feeder cloud did not play a crucial role for precipitation enhancement, as a significant fraction of the hydrometeor mass sublimated before reaching the feeder cloud. However, in other cases of the RACLETS campaign, we found that orographically-induced low-level feeder clouds could enhance precipitation through the seeder-feeder mechanism and provide an environment for secondary ice production mechanisms (Ramelli et al., 2020b)."*

11) *Figure 9. Is it likely that the TKE measured close to the surface at Gotschnagrat is representative of the turbulence in the atmosphere? Isn't it more likely to be due to local shear close to surface? What are the wind speeds at this location? The direction is the same as at the surface, but Fig. 5a suggests high wind speeds at this height, is the increase in TKE just due to an increase in wind speed close to the surface? Why would this then correlate with the precipitation rate elsewhere in the valley? How does this relate to the conceptual figure?*

Thank you for pointing this out. It is likely that the high TKE values measured by the 3D sonic anemometer were related to the high wind speed close to the surface and not representative of the turbulence in the atmosphere. Accordingly, we have removed the TKE observations in the revised manuscript.

12) *I'm not convinced of the usefulness of any the correlation coefficients described here. How do they relate to any expected dynamical or microphysical processes? The shear layer appears to coincide with the sublimation zone. For this case study, any attempt to link the surface precipitation to the maximum reflectivity in the profile should at least take the varying sublimation depth into account.*

Thank you for the comment. We agree that it is difficult to relate these coefficients to the dynamical and microphysical processes. We removed Appendix B and the discussions related to the correlation coefficients in Section 4.2 in the revised manuscript. Instead, we focus on the relationships between the vertical velocity and the low-level cloud properties (see also response to review comment #8).

**Technical comments**

13) *Line 52: Is this wind shear value for wind shear in the horizontal or in the vertical?*

Thank you for this comment. We changed it to "vertical shear in the horizontal wind" in the revised manuscript (page 2, line 53).

14) *Lines 91-92: Do you mean 'the mean ridge height'?*

Thank you for this comment. We changed it to "mean ridge height" (page 4, line 94-95).

15) *Line 116,118: Isn't Vaisala a Finnish company?*

Thank you for pointing this out. We change it in the revised manuscript.

16) *Figure 2 caption: This should state 'taken by the Meteosat 2nd Generation (MSG) satellite'.*

Thank you for pointing this out. We exchanged the satellite picture with a reanalysis plot of the synoptic situation in the revised manuscript.

17) *Line 228: Cloud base? The ice cloud continues to the surface during P1 and P2. Since all ice is falling (precipitating), the ice cloud base is defined in terms of visibility, not in terms of relative humidity (changes in growth or evaporation rate). Hence it is only during P3 that there is an ice cloud base.*

Thank you for pointing this out. We redefined it as onset/beginning of the sublimation layer in the revised manuscript.

---

## Author Response (AR2)

Reviewer comments on 'Influence of low-level blocking and turbulence on the microphysics of a mixedphase cloud in an inner-Alpine valley' by Fabiola Ramelli et al.

Second Response to Reviewer #2

We would like to thank the anonymous referee for his/her constructive and helpful feedback and suggestions on the manuscript. We incorporated the suggestions within the revised manuscript, which significantly improved the quality of the manuscript. In the following, we will address the reviewer's comments and present our responses and changes in the revised manuscript. Reviewer comments are reproduced in blue and the author responses are in black. All line numbers in the author's response refer to the revised manuscript.

**General comments**

1) The authors have responded to all review comments satisfactorily. The only issue noted is the wording with respect to the description of what physical process is responsible for the shear layer, and what physical processes the shear layer is responsible for. There are also have some minor comments (mostly technical) which should also be possible to resolve rapidly without need for a further review.

Thank you for your helpful comments. Following your suggestions, we changed the description of the physical processes related to the shear layer (see responses to specific comments).

**Specific (mostly technical) comments**

- 2) Line 17: Suggest you rephrase this to '.. in this particular case (since the majority of the precipitation evaporated when falling through the subsaturated layer above), ..'
  Thank you for this comment. This was changed in the revised manuscript (page 1, line 18).
- 3) Line 47: Suggest you rephrase this to 'In this study ..' Thank you for this comment. This was changed in the revised manuscript (page 2, line 49).
- 4) Line 63: Suggest you rephrase this to 'This work ...' Thank you for this comment. This was changed in the revised manuscript (page 3, line 65).
- 5) Line 96: Suggest you rephrase this to 'During the case study presented here, the large-scale flow was from the south-western direction..'
  Thank you for this comment. This was changed in the revised manuscript (page 4, line 98-99).
- 6) Line 121: The radar wind profiler also uses DBS to obtain the horizontal wind so is not vertically-pointing only. Typical operation may be 3-5 beams, and the elevation angle is usually between 66 and 76 degrees from horizontal.

Line 124: DBS is short for Doppler Beam Swinging.

*Line 125: I assume 'an elevation angle of 75 degrees from horizontal'?*

Thank you for pointing this out. Indeed, the wind profiler also operated in DBS mode. We added the following sentence, changed the definition of DBS and specified the elevation angle in the revised manuscript (page 5, line 127-128): *"Both wind profiler and wind lidar operated in Doppler Beam*

Swinging (DBS) mode with one vertical and four oblique beams at an elevation angle of 75° from horizontal."

7) Line 138: Suggest you rephrase this to 'The observations presented in this case study were measured in a ..'.

Thank you for this comment. This was changed in the revised manuscript (page 7, line 142).

8) Line 166: You state the dielectric factor here |K|^2, which is 0.93 for liquid water drops at 0 C at X-band. However, it is not quite constant with radar frequency (0.934 at S-band, 0.881 at Ka-band, and 0.686 at W-band).

Thank you for pointing this out. We specified the dielectric factor in the Ka-band radar frequency band in the revised manuscript (page 7, line 171-172): *"Liquid water has a higher refractive index (0.88 for Ka-band) than ice (0.197) (Houze Jr, 2014)."*

9) Line 176: There is unlikely to be any liquid water droplets below the top liquid layer during the period when the low-level cloud was not present (17:45-18:40 UTC) as this would be visible in the lidar backscatter and depolarization fields during this time period. It is likely that there is a liquid layer at the top of the mid-level cloud throughout the time period but this is difficult to state definitively. There may be liquid water present within (rather than only at the top of) the mid-level cloud at other periods but there is no evidence for this. The LWP measurement values cannot easily be used to diagnose whether there are multiple layers either. I suggest removing the sentences starting on line 176.

Thank you for this comment. Following your suggestion, we remove the sentence regarding the presence of liquid water below 3500 m. Furthermore, we rephrased the sentence about the LWP measured by the microwave radiometer (page 9, line 181-182): "Indeed, the LWP measured by the microwave radiometer ranged between 20 g  $m^{-2}$  and 100 g  $m^{-2}$  during the measurement period (Fig. 4c), suggesting the presence of liquid water in some regions (likely at the top) of the mid-level cloud."

10) Figures 5 and 7. Suggest stating '.. horizontal wind speed and ..'

Thank you for this comment. This was changed in the revised manuscript.

**11) Line 228: Suggest 'The observations presented here ..'**

Thank you for this comment. This was changed in the revised manuscript (page 14, line 233).

12) Lines 266-267 and 384: The shear is very unlikely to be responsible for creating an ice supersaturated environment. The difference in relative humidity is likely to be due to there being two atmospheric layers with different air motion and the shear results from the difference in the wind speeds for these two atmospheric layers.

Thank you for this comment. We agree that the difference in wind speed between the blocked layer and the cross-barrier flow aloft is responsible for the strong vertical wind shear. Enhanced turbulence within this region of strong vertical wind shear can lead to additional updrafts that help producing supersaturation and enhancing hydrometeor growth (e.g., through riming, aggregation). We changed the sentence in Lines 266-267 in the revised manuscript (page 16, line 270-273): *"The spatial coincidence between maximum radar reflectivity, shear layer and increase in LDR was also observed for other fallstreaks (Fig. 3d), suggesting that the enhanced turbulence at the interface between the*  blocked layer in the valley and the upper cross-barrier flow led to additional updrafts that helped produce supersaturation and enhance hydrometeor growth (e.g., through riming, aggregation)." The sentence in line 384 now reads as follows (page 23, line 382-387): "The interface between the blocked layer in the valley and the stronger cross-barrier flow aloft was characterized by a region of enhanced turbulence and vertical wind shear. We found that the region of strong vertical wind shear, the extent of the subsaturated layer within the blocked layer in the valley and as a consequence the amount of precipitation reaching the valley were determined by the strength of the downward propagating cross-barrier flow on the upstream mountain barrier B1 and the strength of the low-level blocking on the windward slope of the downstream mountain barrier B2 (Sect. 4.2)."

13) Line 327: The seeder-feeder mechanism was not seen to operate here at all, because there was so little precipitation falling through from above.

Thank you for this comment. We removed this sentence.

14) Line 328: Suggest 'We assume that in the case study presented here ..'

Thank you for this comment. This was changed in the revised manuscript (page 20, line 332).

- 15) Line 388: Suggest 'The observations presented here were obtained within a region ..' Thank you for this comment. This was changed in the revised manuscript (page 23, line 304).
- 16) Line 397: The shear layer is not really a 'separate' layer, it describes the boundary between the in-valley blocked flow and the stronger cross-barrier flow aloft. I.e. enhanced turbulence is present due to there being a strong gradient (shear) in the vertical profile of the horizontal wind. Suggest '.. and the presence of enhanced turbulence in the region of strong vertical wind shear in the boundary between the blocked layer in the valley and the stronger cross-barrier flow aloft.

Thank you for this comment. Following your suggestion, we rephrased the sentence as follows (page 24, line 403-405): *"The wind observations indicated the transition from a blocked to an unblocked low-level flow during the observational period and the presence of enhanced turbulence in the region of strong vertical wind shear in the boundary between the blocked layer in the valley and the stronger cross-barrier flow aloft."*

In addition, we also changed the sentence in the abstract accordingly (page 1, line 7-8): "During this event, the boundary layer was characterized by a blocked low-level flow and enhanced turbulence in the region of strong vertical wind shear at the boundary between the blocked layer in the valley and the stronger cross-barrier flow aloft".

17) Line 411: This statement is the wrong way round. The sublimation layer arises because the upper crossbarrier flow and lower in-valley flow have different humidities, together with the presence of shear because the horizontal winds also differ.

*Line 412: As in the statement above, precipitation reaches the surface when the subsaturated layer is shallower, not because the shear layer extends to lower altitudes.*

Thank you for these comments. We agree that the subsaturated layer is associated with the different humidities of the blocked layer and the upper cross-barrier flow and that the amount of precipitation in the valley was determined by the vertical extent of the subsaturated layer. We changed the paragraph as follows (page 24, line 416-422): *"The region of enhanced turbulence at the boundary between the blocked layer in the valley and the cross-barrier flow aloft was determined by a complex*

interplay of upstream and downstream effects. More specifically, the region of strong vertical wind shear was observed to lower as stronger cross-barrier flow propagated downward and the low-level blocking weakened on the windward slope of the downstream mountain barrier. Due to the lower humidity of the blocked layer in the valley, the majority of the hydrometeors that formed in the mid-level cloud sublimated when falling through the subsaturated layer. Accordingly, precipitation was only observed in the valley when the subsaturated layer was shallowest."

18) Line 416: It is unlikely that it is the altitude of the shear layer that is the major physical reason responsible, rather it is the altitude at which the subsaturated layer begins. The correlation you suggest arises because the altitude of the shear layer corresponds to the altitude of the top of the subsaturated layer.

Thank you for this comment. Following your suggestion, we changed the sentence in the revised manuscript (page 24, line 422-424): "Thus, we propose that the amount of precipitation observed in a mountain valley is influenced by several factors such as (1) the strength of the cross-barrier flow and low-level blocking, (2) the vertical extent of the subsaturated layer and (3) the thermodynamic state of the boundary layer."

19) Line 424: Do you mean a 'dry layer aloft' here? It's unlikely to be a 'boundary layer aloft'.

Thank you for this comment. We changed it to 'due to the subsaturated layer aloft' in the revised manuscript (page 24, line 431).